# Trajectory-Aware Heuristic Learning for Combinatorial Search

**Mustafa Seddiqi** [1]   **Marta Kersten-Oertel** [1]   **Tiberiu Popa** [1]

## Abstract

Learning effective value heuristics for combinatorial search is difficult, as prior methods rely on surrogate supervision or costly downstream search to assess progress. We introduce a trajectory-aware probabilistic framework that models uncertainty in cost-to-go labels instead of treating them as fixed targets. Heuristic learning is cast as inference over state trajectories using an HMM-style model, where estimated depth-change dynamics define transitions and forward-backward inference yields soft supervision. To evaluate heuristic quality without search, we propose a large-scale local ranking metric that measures a model's ability to order neighboring states. On the Rubik's Cube, our approach consistently improves local ranking accuracy and downstream search performance under matched computational budgets.

## 1. Introduction

Learning effective heuristics for combinatorial search remains a challenge at the intersection of artificial intelligence, optimization, and planning. Learning-based methods have demonstrated that neural networks trained from self-generated data can solve large-scale search problems, such as the Rubik's Cube, at or near optimality (Shen et al., 2020). For example, DeepCubeA learns cost-to-go/value heuristics used within Weighted A search and generalizes across combinatorial puzzles (Agostinelli et al., 2019), and earlier approaches used learned policy/value networks with Monte Carlo Tree Search (MCTS) (McAleer et al., 2019). Other work abandons explicit value learning in favor of move prediction for beam-search-based solvers (Takano, 2023). In most prior work, heuristic quality is assessed only after embedding the model into a particular search algorithm, such as MCTS, weighted A, or beam search.

Rather than relying on direct depth or cost-to-go regression (Imai & Kishimoto, 2011; Yonetani et al., 2021; Tamar et al., 2016) or search-in-the-loop training (Silver et al., 2017; Schrittwieser et al., 2020), we shape the learning signal using a combination of local consistency, trajectory-level structure, and distributional regularization. These signals encourage the heuristic to behave coherently across neighboring states and along trajectories, without committing to exact depth values or a specific search algorithm during training. We train heuristics from randomly generated trajectories using structured depth supervision, and guide both the sampling strategy and training signal using a separate, search-free diagnostic of heuristic quality. When exact solution depths are available, heuristic behavior can be evaluated directly by testing whether it correctly orders nearby states by true distance to the goal. To support this evaluation, we introduce *CubeLocalRank*, a dataset of local state neighborhoods that serves as a proxy for downstream search performance and enables efficient, controlled development of sampling and training choices.

We evaluate the learned heuristics through downstream search and show that, under fixed computational budgets, our approach achieves state-of-the-art performance on established benchmarks. Importantly, these gains are obtained without increasing search effort, indicating that improvements stem from the learned heuristic itself rather than changes in search configuration. Our contributions are threefold: (i) a heuristic learning framework that combines trajectory-based sampling with structured supervision enforcing local and global consistency, without search-in-the-loop training; (ii) *CubeLocalRank*, a publicly released search-free diagnostic dataset for directly evaluating heuristic ordering behavior and guiding the design of sampling and training strategies[1]; and (iii) empirical results showing that heuristics learned with this framework achieve state-of-the-art downstream search performance under equal computational budgets and transfer effectively to other puzzle domains.

**Conflict of Interest Disclosure.** The authors declare no financial conflicts of interest related to this work.

---

[1]Department of Computer Science and Software Engineering, Concordia University, Montreal, Canada. Correspondence to: Mustafa Seddiqi <mustafa.seddiqi@concordia.ca>.

*Proceedings of the 43rd International Conference on Machine Learning*, Seoul, South Korea. PMLR 306, 2026. Copyright 2026 by the author(s).

---

[1]Code, trained models, and the CubeLocalRank benchmark are publicly available at `trajectory-aware-heuristics`.

## 2. Related Work

The Rubik's Cube is a benchmark for learning-guided combinatorial search due to its large state space, well-understood group structure, and availability of optimal solvers (Loukhaoukha et al., 2012; Chen & Lee, 2014; Gower et al., 2024; Joyner, 2008; Cornock, 2015; Meinz et al., 2023; Gromov, 1993). It can be modeled as a shortest-path problem on a discrete state graph (Pohl, 1970), where cube configurations are states, legal face rotations are transitions, and the solved configuration is the goal. This formulation leads to different notions of distance that are used throughout the paper. *Step* denotes the number of random moves applied starting from the solved cube, while *depth* denotes the optimal number of moves required to return a state to the solved configuration. By construction, step $\geq$ depth, since random moves need not follow a shortest path. We refer to a *trajectory* as a sequence of states obtained by applying successive moves starting from the solved configuration.

The *depth* is defined with respect to a chosen move convention of which there are two: (1) the Quarter-Turn Metric (QTM), a $90°$ face rotation counts as one move and a $180°$ rotation counts as two moves, and (2) the Half-Turn Metric (HTM), where both $90°$ and $180°$ rotations are counted as single moves. For the $3 \times 3 \times 3$ cube, the maximum optimal depth is 26 under QTM and 20 under HTM. Unless stated otherwise, depths in this work are reported under HTM solely for computational reasons.

Classical approaches frame cube solving as shortest-path search with admissible heuristics (Hart et al., 1968; Richter et al., 2010; Yoon et al., 2008). Iterative Deepening A* (IDA*) combined with pattern databases (PDBs) remains a gold standard, leveraging precomputed abstractions to produce tight lower bounds on the cost-to-go (Korf, 1985; 1997; Culberson & Schaeffer, 1998; Korf & Felner, 2002; Felner et al., 2004). Related group-theoretic pipelines, most notably Kociemba's two-phase algorithm (Kociemba, 2014), achieve exceptional performance through carefully engineered decompositions of the cube group. While these methods provide optimality guarantees, they depend heavily on domain-specific insight and substantial memory, and do not improve through data-driven learning.

Learning-based approaches seek to replace or augment hand-designed heuristics with neural approximations trained from self-generated experience (Samuel, 1959). DeepCube (McAleer et al., 2019) introduced Autodidactic Iteration (ADI), in which states are generated by reversing random scrambles from the goal and supervised using shallow search backups. A joint value–policy network trained in this manner is combined with MCTS at inference, enabling $100\%$ solvability on $3 \times 3 \times 3$ scrambled cube. However, the reliance on MCTS introduces significant inference cost under the cube's high branching factor, and the learned heuristic provides no formal guarantees of optimality.

DeepCubeA (Agostinelli et al., 2019) simplified this pipeline by eliminating explicit policy learning and MCTS. Instead, it trains a value-only network using deep approximate value iteration (DAVI) and uses the learned heuristic within Batch Weighted A* (BWAS) (Ebendt & Drechsler, 2009). Goal-rooted sampling induces a natural curriculum, while batch expansion amortizes neural evaluation across many frontier nodes. This approach achieves $100\%$ solvability with $60.3\%$ optimal solutions on the DeepCubeA 1k benchmark, and transfers easily to other domains such as sliding-tile puzzles, Lights Out, and Sokoban—highlighting the generality of value-only supervision paired with wide best-first search.

Subsequent work has further reduced the strength of the supervision signal. EfficientCube (Takano, 2023) abandons value prediction entirely, instead learning inverse dynamics: given a scrambled state, the network predicts the last random move applied. This policy-only signal biases search toward actions that undo scrambling. Inference is performed using beam search rather than A*-style algorithms, yielding improved speed–optimality trade-offs and achieving $69.6\%$ optimality at beam width $2^{18}$. CayleyPy (Chervov et al., 2025) builds on EfficientCube using agents and random moves as labels. Despite its simplicity, this approach achieves state-of-the-art performance on the DeepCubeA 1k benchmark, reaching $75.4\%$ optimality at beam $2^{18}$ and $97.3\%$ at beam $2^{24}$.

A challenge shared by learning-based solvers is the lack of large-scale, purely supervised training data with exact optimal distances. For cubes larger than $3 \times 3 \times 3$, no optimal solvers are known, and even for the $3 \times 3 \times 3$ cube, computing optimal solutions remains too slow to generate datasets proportional to the state space. As a result, most methods rely on goal-rooted random walks to generate training data, producing a *step index s* equal to the number of random moves applied from the goal. However, learning objectives are defined in terms of *optimal depth d*, the shortest-path distance to the goal, and random walks do not generally follow shortest paths. This yields noisy labels with $s \geq d$, with exact depth information available only up to a small reference depth $d_{\text{ref}}$.

Existing approaches handle this mismatch in different ways. CayleyPy uses the step index as a noisy proxy for depth and relies on wide beam search to compensate for labeling errors. DeepCubeA avoids explicit step-based supervision altogether, instead enforcing local consistency through iterative bootstrapping between neighboring states. In contrast, our work explicitly models the relationship between random-walk steps and optimal depth by introducing a lightweight hidden Markov model (HMM) for depth evolution along

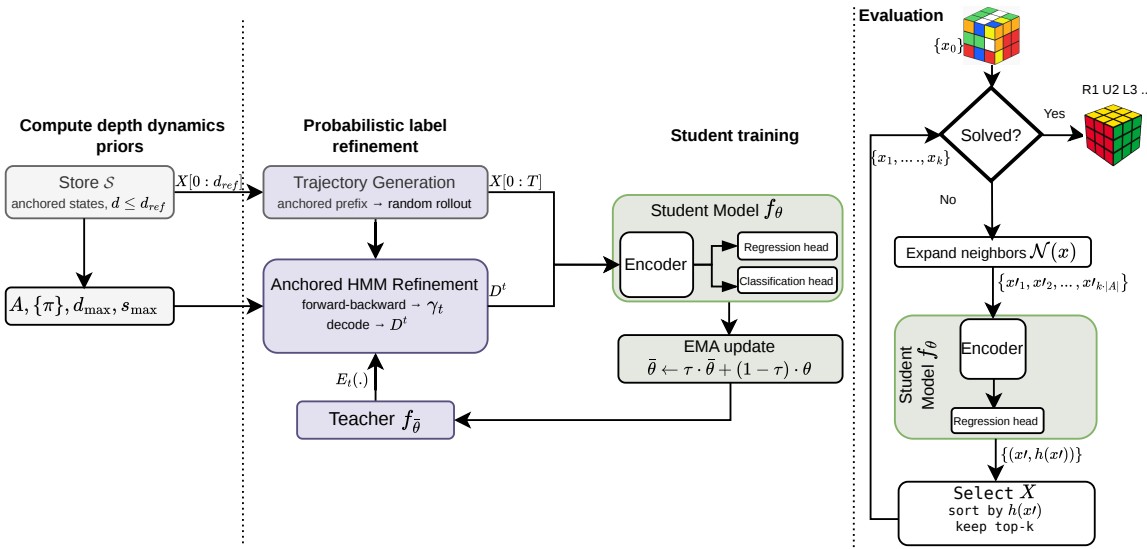

Figure 1. **Framework overview.** We first compute a fixed depth-dynamics prior from shallow exact-depth states. Training then couples label refinement and optimization: anchored trajectories are extended by random rollout, an EMA teacher supplies HMM emissions, the refined labels train the student, and the updated student refreshes the teacher by EMA. At evaluation, the learned regression head guides beam search.

trajectories, using standard forward–backward inference for posterior refinement (Rabiner, 1989).

## 3. Methodology

### 3.1. Overview

Learning value heuristics for large combinatorial puzzles requires depth supervision at a scale that exact solvers cannot provide. Random walks from the goal yield a step index $s$, which upper-bounds the true optimal depth $d$ but becomes increasingly noisy as trajectories grow longer. For Rubik's Cube, exact depths are available only for shallow states, leaving most training data without reliable depth labels.

Our approach is based on two observations. First, optimal depth evolves smoothly along trajectories and is constrained by local puzzle geometry, making it natural to treat depth as a latent trajectory-level variable rather than as an independent label per state. Second, this structure enables noisy predictions to be corrected using sparse exact anchors and simple consistency constraints, without invoking search during label inference.

Together, these observations enable probabilistic depth labeling at scale. Building on this perspective, we propose a framework for learning and evaluating heuristics for combinatorial search based on local ordering consistency among neighboring states. The framework consists of two aligned

components: (i) a trajectory-based depth inference procedure that refines noisy supervision, and (ii) a search-free evaluation protocol that measures heuristic quality through local ordering relations, with search used only for downstream validation.

The framework, illustrated in Figure 1, consists of four stages: (1) *Computing a depth dynamics prior*, to estimate a depth-change model and a step-conditioned depth prior; (2) *Probabilistic label refinement*, in which an anchored HMM combines sparse exact depths, model predictions, and structural constraints to infer consistent depth labels; (3) *Heuristic training*, where a neural network is trained on the refined labels using supervised and structural losses; and (4) *Evaluation*, where the learned heuristic is assessed both using a search-free local ranking benchmark and through downstream search.

We demonstrate this framework on the Rubik's Cube using *CubeLocalRank*, a solver-labeled dataset of local neighborhoods constructed under the HTM. While CubeLocalRank provides a concrete instantiation, the framework itself is general and does not depend on this specific domain or metric. The following sections describe each stage in detail and present the resulting empirical performance.

## 3.2. Depth Dynamics Prior

We model local depth evolution under a single random move using a categorical depth-change distribution $q_\Delta(d) = P(\Delta \mid D = d)$, with $\Delta \in \{-1, 0, +1\}$, capturing whether a move tends to decrease, preserve, or increase optimal depth at depth $d$.

These dynamics are computed for depths $d \leq d_{\text{ref}}$ using exactly labeled states. We create a store $\mathcal{S}$ of exactly labeled states for all depths $d \leq d_{\text{ref}}$, constructed once via breadth-first search. We extend $q_\Delta(d)$ beyond $d_{\text{ref}}$ using a simple stationary-tail approximation. The finite-depth prior construction, including the tail behavior and truncation used to determine the effective modeling range, is described in Appendix A.2.

The resulting depth-change model induces a Markov transition matrix $A$ over discrete depths and a family of step-conditioned priors $\pi_t(d)$ obtained by propagation from the solved state. Together, $(A, \{\pi_t\})$ encode the expected geometric evolution of depth under random walks, independently of any learned heuristic. Algorithm 1 summarizes how $(A, \{\pi_t\})$ are computed.

One subtle challenge is choosing the provisional modeling cap $d_{\text{cap}}$ used to construct the finite prior. When the true domain diameter is known, it provides a natural choice; otherwise $d_{\text{cap}}$ can be set as a conservative upper modeling range. In either case, $d_{\text{cap}}$ is not the final output depth range. We infer the effective depth limit $d_{\text{max}}$ and trajectory horizon $s_{\text{max}}$ from the induced tail behavior of the prior, and the HMM is then constructed on the resulting finite depth space. Thus, $d_{\text{cap}}$ controls prior construction, while $(d_{\text{max}}, s_{\text{max}})$ define the actual refinement range used during training.

## 3.3. Probabilistic Label Refinement via Anchored HMM

**Trajectory construction.** To train our model trajectories are generated by augmenting the data in the store $\mathcal{S}$ by sampling seed states at depth $d_{\text{ref}}$ and extending each seed forward by random moves for $s_{\text{max}} - d_{\text{ref}}$ additional steps. This produces trajectories $\{X_t\}_{t=0}^{s_{\text{max}}}$ in which the initial segment is exactly labeled and the remainder has unknown depth. Crucially, trajectories are grown by single-step continuation rather than repeated restarts from the goal, preserving temporal coherence for inference.

**Anchors and emissions.** For each trajectory position $t > d_{\text{ref}}$ in our dataset, we attempt to match the state against the labeled store. If a match is found at depth $d_m$, we impose a hard anchor by fixing $D_t = d_m$. For non-anchored positions, we obtain depth likelihoods from a teacher network with two outputs: a depth-classification head and a regression head predicting a scalar depth. The regression output is converted into a discrete likelihood over depths and

combined with the classification score to form emissions $E_t(d)$. Hard constraints induced by anchors, depth caps, and step feasibility are enforced by masking invalid depths.

**Anchored HMM inference.** Given emissions $\{E_t(d)\}$ and the depth transition matrix $A$ induced by the depth dynamics prior, we perform forward–backward inference to compute posterior marginals

$$\gamma_t(d) = p(D_t = d \mid X_{0:s_{\text{max}}}),$$

along each trajectory. This yields smoothed depth distributions that reconcile local predictions with global consistency constraints. Finally, pseudo-labels are obtained by per-step marginal decoding $\hat{D}_t = \arg\max_d \gamma_t(d)$. Algorithm 2 summarizes the full refinement procedure, and Figure 2 visualizes the refinement step: the step-conditioned prior, anchored emissions, and posterior marginals.

## 3.4. Heuristic Training

Anchored trajectories are first constructed using a small store of exactly labeled states (§3.2). An EMA teacher then provides emissions for anchored HMM refinement, yielding refined depth labels along each trajectory. Finally, a student model is trained on these labels using supervised and structural losses, and its parameters are fed back to the teacher via EMA.

**Architecture.** The student model consists of a shared encoder with two output heads: a scalar depth regression and a discrete depth classification. The encoder follows the CayleyPy backbone to enable direct comparison, including the same state encoding, hidden widths, and residual structure. Given a state $x$, the regression head predicts a scalar depth $\hat{d}(x)$, while the classification head outputs logits over depths $\{0, \ldots, d_{\text{max}}\}$, inducing $p_\theta(d \mid x) = \text{softmax}(\ell_\theta(x))_d$. The dual-head design supports both scalar supervision and distributional regularization.

**Refinement–training loop.** Training uses an EMA teacher $f_{\bar{\theta}}$ to produce emissions for anchored HMM refinement. For each batch, we sample anchored trajectories, run forward–backward inference using the fixed prior $(A, \{\pi_t\})$, decode pseudo-labels $\hat{D}_t = \arg\max_d \gamma_t(d)$, and update the student $f_\theta$ on the resulting state-label pairs. The teacher is then updated by EMA,

$$\bar{\theta} \leftarrow \tau\bar{\theta} + (1 - \tau)\theta.$$

This couples refinement and optimization: improved student predictions lead to better teacher emissions in later rounds.

**Training objective.** The student is trained with

$$\mathcal{L} = \lambda_{\text{reg}}\, \mathcal{L}_{\text{MSE}} + \lambda_{\text{cls}}\, \mathcal{L}_{\text{CE}} + \lambda_{\text{nbr}}\, \mathcal{L}_{\text{nbr}}$$
$$+ \lambda_{\text{step}}\, \mathcal{L}_{\text{step}} + \lambda_{\text{glob}}\, \mathcal{L}_{\text{glob}} + \lambda_{\text{inv}}\, \mathcal{L}_{\text{inv}}.$$

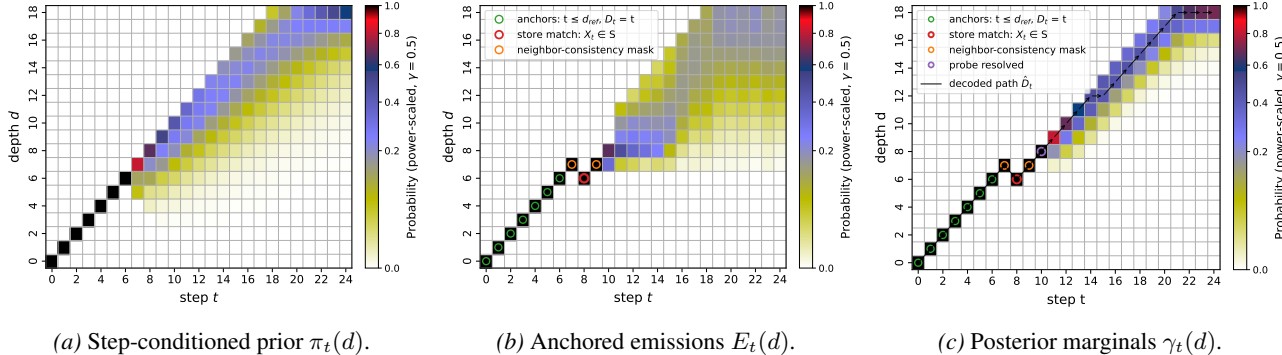

*(a)* Step-conditioned prior $\pi_t(d)$.  *(b)* Anchored emissions $E_t(d)$.  *(c)* Posterior marginals $\gamma_t(d)$.

*Figure 2.* **Anchored HMM depth refinement.** (a) The step-conditioned prior $\pi_t(d)$ describes where random-walk trajectories are expected to lie in depth–time space. (b) Emissions $E_t(d)$ combine exact anchors, teacher predictions, and feasibility constraints. (c) Forward–backward inference produces posterior marginals $\gamma_t(d)$, which give smoothed pseudo-labels along the trajectory.

The regression and classification losses train the scalar and distributional heads on decoded pseudo-labels using normalized sample weights. Neighbor consistency penalizes large prediction jumps between adjacent trajectory states. The KL and feasibility terms regularize the classification head so that it provides stable emissions for later HMM refinement. Definitions of the loss terms and teacher-emission construction are given in Appendix C.

### 3.5. Evaluation: *CubeLocalRank*

Evaluating heuristics solely through downstream search can be misleading: highly optimized search procedures, especially beam search with large beam sizes, may reduce or mask differences in heuristic quality. Downstream search is also computationally expensive, making rapid evaluation and iteration difficult. We therefore introduce *CubeLocalRank*, a search-free evaluation protocol based on local ranking consistency, together with a benchmark dataset that instantiates this protocol on the Rubik's Cube.

CubeLocalRank is constructed on the $3 \times 3 \times 3$ Rubik's Cube under the Half-Turn Metric (HTM). It contains approximately 167k reference states with exact optimal depths, each paired with exact-depth local neighborhoods. These neighborhoods form a static benchmark for large-scale evaluation of heuristic quality without invoking a downstream search procedure. Full construction details, dataset statistics, and metric definitions are provided in Appendix D.

Given a reference state $x$, CubeLocalRank evaluates whether a heuristic correctly prioritizes the best candidate within a local neighborhood $\mathcal{N}(x)$. Candidates are ordered by predicted depth and compared against their true optimal depths. In the main paper, we report *Aggregate Rank@1*, which summarizes performance cumulatively up to depth $d$; in the appendix, we additionally report the per-depth *Rank@1* breakdown. This directly measures the quality of the local expansion decisions used by greedy, beam, and $A^*$

search, without running the full search procedure.

Although CubeLocalRank is instantiated on the Rubik's Cube, the evaluation protocol itself is more general and can be applied to other combinatorial search domains whenever exact local depth supervision is available.

## 4. Results

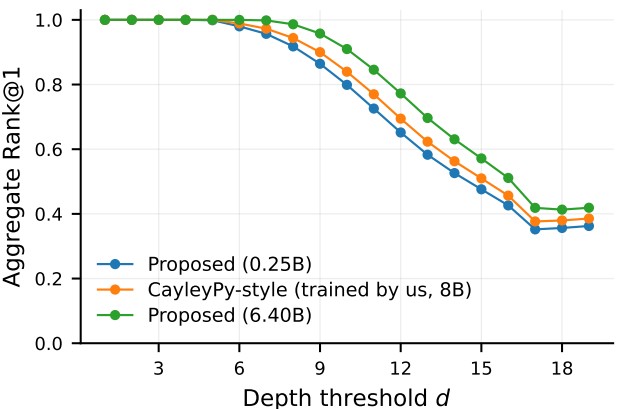

*Figure 3.* Comparison of Aggregate Rank@1 on CubeLocalRank (HTM, level-1) as the evaluation depth threshold $d$ increases.

We evaluate heuristic quality using both direct (search-free) metrics and downstream search performance. Results are reported on the CubeLocalRank and DeepCubeA benchmarks for the $3 \times 3 \times 3$ Rubik's Cube under the HTM and QTM metrics. CubeLocalRank provides a search-free evaluation of local decision quality by measuring whether heuristics correctly rank shallow neighbors, while DeepCubeA evaluates performance in full downstream search. We further demonstrate scalability on larger puzzle domains from the Santa Claus 2023 benchmark suite, including the $4 \times 4 \times 4$ and $5 \times 5 \times 5$ Rubik's Cubes and the $6 \times 10$ Globe puzzle (Holbrook et al., 2023).

*Table 1.* **Main results on the DeepCubeA benchmark.**

| Setting | Method | Params (M) | Train size ($\times 10^9$) | Search | Avg. len. | Optimality (%) |
|---|---|---|---|---|---|---|
| $3 \times 3 \times 3$ **– HTM** | | | | | | |
| HTM | CayleyPy-style (trained by us) | 4.0 | 10 | Beam ($2^{18}$) | 18.154 | 59.6 |
| HTM | Ours | 4.0 | 10 | Beam ($2^{18}$) | 18.100 | 63.3 |
| HTM | CayleyPy-style (trained by us) | 4.0 | 52 | Beam ($2^{18}$) | 18.110 | 62.3 |
| HTM | Ours | 4.0 | 52 | Beam ($2^{18}$) | 18.040 | **68.1** |
| $3 \times 3 \times 3$ **– QTM** | | | | | | |
| QTM | DeepCubeA | 14.7 | 10 | Batch A$^*$ ($\lambda$=0.6) | 21.500 | 60.3 |
| QTM | Ours | 4.0 | 22 | Batch A$^*$ ($w$=0.8) | 20.930 | 85.4 |
| QTM | EfficientCube | 14.7 | 52 | Beam ($2^{18}$) | 21.260 | 69.8 |
| QTM | CayleyPy (reported/tested) | 4.0 | 8–52 | Beam ($2^{18}$) | 21.140 | 75.4 |
| QTM | Ours | 4.0 | 10 | Beam ($2^{18}$) | 21.151 | 77.8 |
| QTM | Ours | 4.0 | 52 | Beam ($2^{18}$) | 21.029 | 80.5 |
| QTM | CayleyPy (reported) | 4.0 | 8 | Beam ($2^{24}$) | 20.690 | 97.3 |
| QTM | CayleyPy ensemble (26$\times$) | 26$\times$4.0 | 26$\times$8 | Beam ($2^{24}$) | 20.670 | 98.4 |
| QTM | Ours | 4.0 | 52 | Beam ($2^{24}$) | 20.683 | **97.7** |

**Notes:** All results are evaluated on the fixed test scrambles released with DeepCubeA for $3 \times 3 \times 3$, using HTM and QTM. For HTM, CayleyPy did not release trained models; we therefore retrain the original CayleyPy architecture following its published procedure for direct comparison. Under QTM, DeepCubeA-style models are evaluated using batch A$^*$, while CayleyPy and EfficientCube use beam search. CayleyPy QTM results at beam $2^{18}$ are nearly invariant across training sizes (8–52B), so we report a single aggregated row.

## 4.1. Heuristic Quality via CubeLocalRank

CubeLocalRank directly evaluates local heuristic quality without running search. On level-1 neighborhoods, our model achieves higher overall Aggregate Rank@1 than the CayleyPy-style baseline trained by us, improving from 0.40 to 0.43. This indicates more reliable local ordering of successor states by true optimal depth. Figure 3 shows the aggregate Rank@1 curve as the evaluation depth threshold increases.

Additional level-1 and level-2 CubeLocalRank results, together with the corresponding downstream beam-search sweeps, are reported in Appendix G.

## 4.2. $3 \times 3 \times 3$ **Rubik's Cube**

Table 1 reports solution quality on the DeepCubeA benchmark under both HTM and QTM. Under HTM, our model consistently outperforms a CayleyPy-style baseline trained under the same protocol and parameter budget. Optimality is measured with respect to HTM solution lengths recomputed for the DeepCubeA test states using an optimal solver (Skalski). At a fixed beam size of $2^{18}$, increasing the training set from 10B to 52B yields a substantial improvement in optimality for our model (63.3% $\rightarrow$ 68.1%), while the corresponding CayleyPy-style model shows only marginal gains. At larger beam widths, our model continues to benefit from stronger heuristic guidance: with beam $2^{23}$, it achieves 93.3% optimality, approaching oracle-level performance. These results indicate that improved local ranking quality translates into more effective search under realistic HTM budgets.

Under QTM, we compare against DeepCubeA, EfficientCube, and reported CayleyPy results. Using beam search at $2^{18}$, our model improves steadily with training scale (77.8% $\rightarrow$ 80.5%), outperforming EfficientCube (69.8%) and CayleyPy (75.4%) at comparable settings. These gains indicate that stronger heuristic guidance leads to more effective search under moderate QTM budgets.

We also evaluate using batch A$^*$ to match DeepCubeA-style evaluation. Our 4.0M-parameter model achieves 85.4% optimality, substantially exceeding DeepCubeA (60.3%) despite using fewer parameters.

At very large beam widths ($2^{24}$), all methods approach near-optimal performance, reflecting saturation of the search procedure. In this regime, our single model achieves 97.7% optimality, closely matching reported CayleyPy performance (97.3%) and approaching that of a 26-model CayleyPy ensemble (98.4%). This convergence highlights the diminishing discriminative power of saturated search accuracy and emphasizes the strength of the learned heuristic itself, rather than reliance on ensembling or extreme search budgets. Full beam-size sweeps for HTM and QTM are reported in Appendix G.

**Search Efficiency.** Across beam budgets, our beam search attains higher throughput than CayleyPy under the same hardware and protocol. On average, we observe a $\sim$1.7$\times$ speedup, e.g., 12.2s vs. 7.14s per instance at beam $2^{18}$ (CayleyPy vs. ours). We use the same stagnation rule via exact store matching, which does not change solvability in our tests and primarily improves throughput. A detailed cost analysis, including training overhead, exact matching, nu-

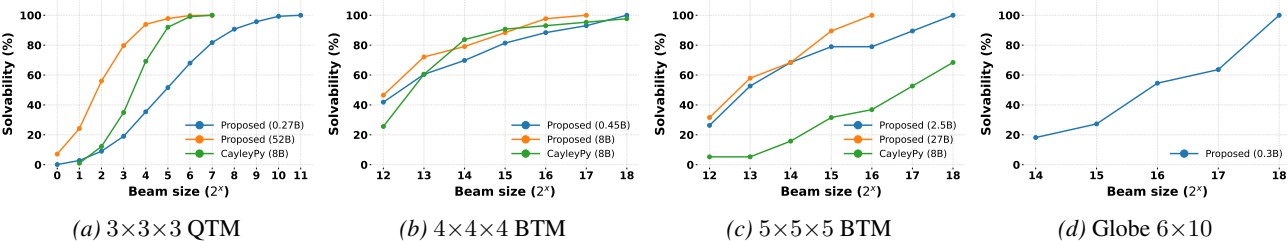

*(a)* $3{\times}3{\times}3$ QTM      *(b)* $4{\times}4{\times}4$ BTM      *(c)* $5{\times}5{\times}5$ BTM      *(d)* Globe $6{\times}10$

*Figure 4.* **Cumulative solvability across beam budgets.** Each panel reports the percentage of test instances solved by beam search as the beam budget increases. For each instance, beam sizes are tried in increasing order, and an instance solved at beam $B$ is counted as solved for all larger beams. The $3{\times}3{\times}3$ QTM panel is evaluated on the 1k DeepCubeA test set with maximum search horizon 100. The $4{\times}4{\times}4$ and $5{\times}5{\times}5$ panels are evaluated on 43 and 19 Santa Claus 2023 instances, respectively, with maximum search horizon 200. The Globe $6{\times}10$ panel uses 11 instances, consisting of 10 generated random scrambles and one Santa Claus 2023 instance, also with maximum search horizon 200. All panels use search-side exact matching with $d_{\mathrm{ref}} = 5$. BTM denotes the block-turn metric.

merical precision, and matched search-budget comparisons, is provided in Appendix E.

### 4.3. Scalability and Generalizability

We evaluate scalability of our method by measuring solvability as a function of beam-search budget across increasingly large permutation puzzles. For $3{\times}3{\times}3$, this provides a complementary view to the solution-quality results in Table 1: rather than measuring solution length at a fixed large beam, it shows how quickly a heuristic begins solving instances as the beam budget increases. For larger puzzles, where optimal solution lengths are unavailable, solvability is the primary evaluation metric.

For each test instance, we sweep beam sizes in increasing order. At each beam size, search is run with a fixed maximum horizon; once a valid solution is found, the instance is counted as solved for that beam and for all larger beams. Figure 4 reports cumulative solvability and summarizes the evaluation settings, showing how solvability changes as a function of beam budget across $3{\times}3{\times}3$ QTM and the larger puzzle domains, together with the corresponding evaluation settings.

On $3{\times}3{\times}3$ under QTM, our 52B model reaches 90% solvability by beam $2^4$ and full solvability by beam $2^7$. The released CayleyPy 8B model also reaches full solvability by beam $2^7$, but requires beam $2^5$ to reach 90%. The smaller 0.27B version of our model reaches 90% by beam $2^8$ and full solvability by beam $2^{11}$. Notably, the 52B QTM model solves 7.1% of instances under greedy search, i.e. beam $2^0 = 1$, while CayleyPy solves none at that beam.

We observe the same scaling trend under HTM using the same cumulative beam-sweep protocol. Scaling training from 0.25B to 52B shifts the solvability curves substantially leftward: the 52B model reaches 90% solvability by beam $2^6$ and full solvability by beam $2^{10}$, compared with $2^9$ and $2^{12}$, respectively, for the 0.25B model. This indicates that the improvement is not specific to QTM, but also appears

under the move convention used for CubeLocalRank and the main HTM experiments.

Taken together, the QTM and HTM results show that on $3{\times}3{\times}3$, where evaluation uses 1000 test states, the learned heuristic improves the beam budget required for high solvability under both QTM and HTM.

On larger cubes, our single-model heuristics remain effective under practical beam budgets. For $4{\times}4{\times}4$ under BTM, our 8B model solves all 43 Santa Claus instances by beam $2^{17}$, while the released CayleyPy 8B model reaches 97.7% solvability at beam $2^{18}$. Our smaller 0.45B model also reaches full solvability by beam $2^{18}$. For $5{\times}5{\times}5$, the difference is larger: our 27B model solves all 19 instances by beam $2^{16}$, and our 2.5B model solves all instances by beam $2^{18}$, whereas the released CayleyPy 8B model reaches 68.4% solvability at beam $2^{18}$. Finally, on Globe $6{\times}10$, our 0.3B model reaches full solvability at the largest evaluated beam, $2^{18}$.

Overall, the same training framework transfers to larger cubes and to a non-cube permutation puzzle. Additional large-puzzle diagnostics, including assumed depth-cap sensitivity, fixed-beam solution-length comparisons, and the limited $6{\times}6{\times}6$ study, are reported in Appendix F.

## 5. Ablation Studies

We ablate the components most directly responsible for heuristic quality: trajectory-level label refinement, neighbor consistency, and the auxiliary objective terms used to support HMM emissions. Table 2 summarizes the corresponding controlled comparisons. Unless stated otherwise, experiments are conducted on the $3{\times}3{\times}3$ Rubik's Cube under HTM and evaluated using CubeLocalRank Level-1 Aggregate Rank@1. Because some diagnostics use different matched settings, comparisons should be made within each block of Table 2. Additional sensitivity studies are reported in Appendix H.

*Table 2.* **Core ablations.** Aggregate Rank@1 is reported on a 0–1 scale. Parenthesized differences are percentage-point changes. Search optimality is measured on the DeepCubeA benchmark under HTM using beam search with beam size $2^{18}$. In the objective block, scalar depth regression is the base supervised loss, and rows report marginal additions to this base.

| Component | Controlled comparison | Train | Agg. Rank@1 | Search opt. |
|---|---|---|---|---|
| *Trajectory-level label refinement* | | | | |
| Anchored HMM | Full anchored HMM vs. step-based supervision + anchors | 4.2B | 0.413 vs. 0.399 (+1.4 pp) | – |
| | Full anchored HMM vs. prior + anchors, uniform emissions | 4.2B | 0.413 vs. 0.404 (+0.9 pp) | – |
| | Full anchored HMM vs. prior-sampled pseudo-labeling, no HMM | 4.2B | 0.413 vs. 0.381 (+3.2 pp) | – |
| *Neighbor consistency and trajectory strategy* | | | | |
| Neighbor consistency | Goal-rooted restarts: with vs. without neighbor consistency | 10B | 0.414 vs. 0.400 (+1.4 pp) | 62.9% vs. 59.6% (+3.3 pp) |
| | Single-step pipeline: full vs. no neighbor consistency | 10B | 0.422 vs. 0.382 (+4.0 pp) | 63.3% vs. 58.1% (+5.2 pp) |
| *Training objective beyond base regression* | | | | |
| Auxiliary structure | Regression + neighbor consistency vs. regression only | 10B | 0.418 vs. 0.399 (+1.9 pp) | – |
| | Full objective vs. regression + neighbor consistency | 10B | 0.419 vs. 0.418 (+0.1 pp) | – |

For goal-rooted restarts with neighbor consistency, each restart state is paired with one true successor before applying the local consistency loss; thus 10B restart samples correspond to 20B effective state evaluations for that loss.

**Trajectory-Level Refinement.** The first block of Table 2 isolates the effect of anchored HMM refinement. The full model improves over step-based supervision with anchors, over an HMM that retains the prior and anchors but uses uniform emissions, and over independent prior-sampled pseudo-labeling without forward–backward inference. These comparisons separate the roles of the three refinement ingredients: exact anchors, the depth-dynamics prior, and learned teacher emissions. The best performance is obtained only when all three are combined through trajectory-level inference, indicating that the gain is not due to the prior or anchors alone.

**Neighbor Consistency.** Neighbor consistency provides the largest structural gain beyond scalar depth regression. With CayleyPy-style goal-rooted restarts, adding the loss improves both CubeLocalRank and downstream HTM search optimality. The effect is stronger under our single-step trajectory construction: removing neighbor consistency reduces Aggregate Rank@1 by 4.0 percentage points and search optimality by 5.2 percentage points. Thus, local consistency is useful even for restart-style data, but it is most effective when applied to temporally connected trajectories, where adjacent states provide natural one-step constraints.

**Objective Terms.** The scalar regression head is the heuristic used for ranking and search, so regression remains the base supervised objective. Adding neighbor consistency gives the main improvement over regression alone.

The remaining classification and KL terms add little direct Rank@1 gain, but they are retained because they stabilize the distributional predictions used as HMM emissions. In other words, neighbor consistency primarily improves the scalar heuristic, whereas the auxiliary distributional terms support the refinement loop.

**Other Structural Choices.** The remaining ablations mainly affect sample efficiency and early training stability. A nontrivial exact-depth store is important: increasing $d_{ref}$ from 0 to 1 gives a clear gain, while deeper stores provide diminishing returns at larger training scales. The provisional depth cap is robust once it is not too small: $d_{cap} = 15$ undercovers the useful range, whereas $d_{cap} = 20$ and 30 behave similarly. Bounded probe refinement improves intermediate-depth ranking, especially on the $d \le 15$ aggregate at medium training scales. Finally, delaying EMA-based emissions stabilizes early training; after roughly 1B samples, different EMA-start choices differ by about 0.002 Aggregate Rank@1, or 0.2 percentage points.

## 6. Discussion

A central motivation of this work is the observation that random-walk supervision provides only weak and noisy information about optimal depth, especially far from the goal. Prior approaches either ignore this noise, rely on iterative bootstrapping, or implicitly correct it through increasingly powerful search. Our results suggest that explicitly mod-

eling depth as a latent trajectory-level variable yields measurable improvements in heuristic fidelity, even when the probabilistic model is simple. The depth-dynamics prior captures coarse geometric constraints of the state space, while anchored inference propagates reliable information from shallow regions into deeper parts of the trajectory. The ablations further show that these gains arise from combining exact anchors, the prior, and learned emissions through forward–backward inference, rather than from any single source of supervision in isolation. Importantly, the benefits of this modeling choice are visible not only in end-to-end search metrics but also in search-free local ranking behavior. This supports the view that the learned heuristics themselves are more consistent and informative, rather than merely better matched to a particular search configuration.

CubeLocalRank is not intended to replace downstream search evaluation, but to complement it. As beam width or search budget increases, many strong heuristics converge to near-identical end-to-end performance, making it difficult to determine improvements or regressions. Local ranking evaluation isolates the decision-making behavior that directly influences node expansion and provides a statistically efficient way to compare heuristics in regimes where search accuracy saturates. While CubeLocalRank is instantiated on the Rubik's Cube, the underlying protocol—evaluating local ordering consistency using exact neighborhoods— applies broadly to combinatorial search domains where limited optimal supervision is available.

The proposed framework relies on several assumptions that may not hold uniformly across all combinatorial domains. The depth-dynamics prior is estimated from shallow exact-depth statistics and extrapolated to deeper regions; while this approximation is empirically robust on the Rubik's Cube and transfers effectively to some puzzles, it may require adaptation in domains with highly irregular geometry or strongly non-local transitions. Anchored HMM inference introduces additional computational overhead during training, although this cost is amortized and does not affect inference-time performance. CubeLocalRank also requires access to solver-labeled local neighborhoods, which may be unavailable or expensive to compute in some domains. In such cases, search-free evaluation may need to rely on approximate or partial supervision, or be replaced by alternative diagnostics. Importantly, the heuristic learning framework itself does not depend on CubeLocalRank and can be applied independently of this evaluation protocol.

Several extensions of this work are promising. Richer trajectory models, such as action-conditional dynamics or higher-order transitions, could further improve label refinement without invoking search. The observed alignment between local ranking quality and downstream performance also suggests the possibility of training heuristics directly against ranking-based objectives, rather than using them solely for evaluation. More broadly, this work argues for a shift in how learned heuristics are trained and evaluated, away from tightly coupled search-in-the-loop pipelines and toward principled supervision and diagnostics that isolate heuristic quality itself. We hope this perspective encourages further work on scalable, interpretable heuristic learning independent of specific search algorithms.

## 7. Conclusion

We introduced a trajectory-aware framework for learning value heuristics in combinatorial search. Instead of treating random-walk step indices as fixed depth labels, the method models optimal depth as a latent trajectory-level variable and refines supervision by combining sparse exact anchors, a depth-dynamics prior, teacher emissions, and forward–backward inference. The resulting labels support heuristic training with supervised and structural objectives, without search-in-the-loop training.

We also introduced CubeLocalRank, a search-free benchmark that evaluates local ordering consistency with respect to true optimal depth. Across Rubik's Cube benchmarks under HTM and QTM, the learned heuristics improve local ranking and downstream search performance under matched computational budgets. Ablations show that the main gains come from trajectory-level refinement and neighbor-consistency regularization, while auxiliary distributional terms support stable HMM emissions. The same framework also transfers to larger cubes and to the Globe puzzle under fixed search budgets.

## Acknowledgements

We acknowledge the support of the Natural Sciences and Engineering Research Council of Canada (NSERC), under funding reference numbers RGPIN-2021-03477 and RGPIN-2017-06722.

## Impact Statement

This work studies learned heuristics for combinatorial search on synthetic puzzle domains. It does not involve human subjects, personal data, or direct deployment in safety-critical settings. Its primary impact is methodological: improving how search heuristics are trained and evaluated. As with other general-purpose optimization methods, any deployment in real-world decision systems would require domain-specific validation, monitoring, and safeguards.

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

# A. Depth Dynamics and Prior Construction

## A.1. Depth-Dynamics Prior Construction

---

**Algorithm 1** Depth-dynamics prior construction and tail-based range selection

---

1: **Input:**
2:    store $\mathcal{S}$ $(d \leq d_{\text{ref}})$
3:    cap $d_{\text{cap}}$
4:    tail fraction $\omega$
5:    threshold $\epsilon$
6: **Output:** $d_{\max}$, $s_{\max}$, transition $A$, priors $\{\pi_t\}_{t=0}^{s_{\max}}$
7:
8: **Build capped depth dynamics**
9: Compute $q_\Delta(d)$ for $d \leq d_{\text{ref}}$ from $\mathcal{S}$
10: Extend $q_\Delta(d)$ naively to $d_{\text{cap}}$ and construct $A^{\text{cap}}$ on $\{0, \ldots, d_{\text{cap}}\}$
11:
12: **Define tail band**
13: $w \leftarrow \lceil \omega d_{\text{cap}} \rceil$
14: $\mathcal{D}_{\text{tail}} \leftarrow \{d_{\text{cap}} - w + 1, \ldots, d_{\text{cap}}\}$
15:
16: **Detect** $(s_{\max}, d_{\max})$ **via first tail violation**
17: Initialize $\pi_0, \rho_0 \leftarrow \delta_0$ {point mass at depth 0}
18: **for** $t = 0, 1, 2, \ldots$ **do**
19:    $\pi_{t+1} \leftarrow \pi_t A^{\text{cap}}$
20:    $\rho_{t+1} \leftarrow \rho_t + \pi_{t+1}$
21:    $d_{\text{viol}} \leftarrow \min\{d \in \mathcal{D}_{\text{tail}} : \rho_{t+1}(d) - \rho_{t+1}(d-1) > \epsilon\}$
22:    **if** $d_{\text{viol}}$ exists **then**
23:        $s_{\max} \leftarrow t + 1$
24:        $d_{\max} \leftarrow d_{\text{viol}}$
25:        **break**
26:    **end if**
27: **end for**
28:
29: **Collapse and finalize prior**
30: Collapse depths $d > d_{\max}$ into bucket $d_{\max}$ and rebuild $A$ on $\{0, \ldots, d_{\max}\}$
31: Recompute $\{\pi_t\}_{t=0}^{s_{\max}}$ under $A$
32: **Return** $d_{\max}, s_{\max}, A, \{\pi_t\}$

---

## A.2. Finite-Depth Prior Construction: Tail Behavior and Truncation

The depth-dynamics prior $(A, \{\pi_t\})$ introduced in Section 3.2 is constructed from empirical depth-change statistics estimated in the exactly labeled region and then restricted to a finite, practically relevant depth range for training. Algorithm 1 first estimates the depth-change distribution $q_\Delta(d)$ for depths $d \leq d_{\text{ref}}$ from the exact-depth store $\mathcal{S}$. Since exact depth statistics are only available up to $d_{\text{ref}}$, we extend these dynamics beyond the exact region using a simple stationary-tail approximation up to a provisional modeling cap $d_{\text{cap}}$. Figure 5 illustrates this construction for the $3 \times 3 \times 3$ cube under HTM.

The provisional cap is useful for constructing the prior, but it should not necessarily be used directly as the final depth range. Even when the true domain diameter is known, the deepest states may receive negligible probability mass under random walks from the solved state. Using the full capped

range would therefore allocate model output classes and training signal to depths that are theoretically possible but rarely encountered under the trajectory distribution used for learning. Instead, we infer an effective depth limit $d_{\max}$ and trajectory horizon $s_{\max}$ from the induced tail behavior of the step-conditioned prior.

Concretely, the capped transition model first induces a sequence of step-conditioned priors by propagation from the solved state. Let $\pi_t(d)$ denote the probability of being at depth $d$ after $t$ random moves under this capped model. Algorithm 1 accumulates these priors over time and monitors a tail band near $d_{\text{cap}}$. The first significant tail violation indicates that probability mass is beginning to interact with the artificial cap, which provides a data-driven criterion for selecting the effective horizon and depth range.

After this range is selected, all depths greater than $d_{\max}$ are collapsed into the terminal depth bucket. The transition matrix $A$ is then reconstructed on the finite depth space $\{0, \ldots, d_{\max}\}$, and the final step-conditioned priors $\{\pi_t\}_{t=0}^{s_{\max}}$ are recomputed under this truncated model. These are the prior objects used by the anchored HMM refinement procedure in Algorithm 2.

This construction separates the provisional modeling cap from the actual training range. The cap $d_{\text{cap}}$ provides a conservative upper range for constructing and diagnosing the prior, while $(d_{\max}, s_{\max})$ define the finite latent depth space and trajectory horizon used during refinement. For the $3 \times 3 \times 3$ HTM setting shown here, the provisional cap is chosen as $d_{\text{cap}} = 20$, matching the known HTM diameter, but the final training range is selected from the prior's tail behavior rather than fixed directly to this cap.

# B. Anchored HMM Inference

## B.1. Anchored HMM Refinement

## B.2. Anchor Handling and Probe Refinement

Algorithm 2 begins by constructing anchored trajectories from the exact-depth store. Lines 9–15 use store membership to recover exact matches up to $d_{\text{ref}}$ and to impose hard anchors whenever a later trajectory state also matches the store. These anchors provide exact supervision at matched positions and constrain the subsequent HMM inference. After emissions are formed for the remaining positions, local step-feasibility masks are applied (line 16) to rule out depth assignments that violate the one-step trajectory structure. Forward–backward inference is then run under the resulting anchor and mask constraints to obtain posterior marginals and decoded pseudo-labels.

**Probe Refinement near the Anchor Boundary.** Exact anchors become sparse immediately beyond the labeled

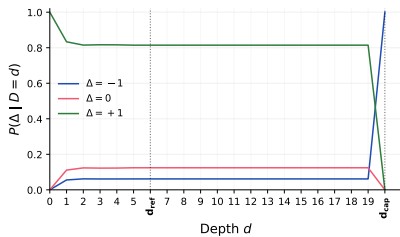 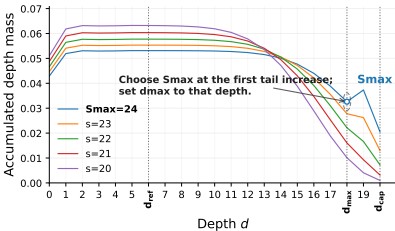 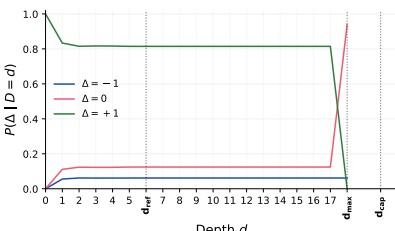

*(a)* Extended depth-change dynamics $q_\Delta(d)$ from the exact regime $d \le d_{\text{ref}}$ to the provisional cap $d_{\text{cap}}$.

*(b)* Tail diagnostics used to identify the effective modeling range $(d_{\text{max}}, s_{\text{max}})$ from cumulative prior occupancy.

*(c)* Final truncated dynamics after collapsing depths beyond $d_{\text{max}}$ into the terminal bucket.

*Figure 5.* **Finite-depth prior construction from a provisional cap.** Illustration on the $3{\times}3{\times}3$ Rubik's Cube under HTM with $d_{\text{ref}} = 6$ and $d_{\text{cap}} = 20$. The tail criterion selects $d_{\text{max}} = 18$ and $s_{\text{max}} = 24$. Starting from empirical depth-change statistics in the exact-depth region, we extend the dynamics to the provisional cap, identify the effective range from tail occupancy, and collapse depths beyond $d_{\text{max}}$ into a terminal bucket.

region $d_{\text{ref}}$. To partially extend their effect without explicitly storing deeper exact-depth sets, Algorithm 2 optionally applies a bounded probe refinement at lines 21–29. For each non-anchored position whose decoded depth satisfies $\hat{D}_t \le d_{\text{ref}} + \Delta_{\text{probe}}$, the state is expanded up to $\Delta_{\text{probe}}$ hops in search of a store match. If a match at depth $d_m$ is found after $h$ hops, the position is hard-anchored as $D_t = d_m + h$ and the refined constraint is injected back into inference.

If no match is found, then depths $d \le d_{\text{ref}} + \Delta_{\text{probe}}$ are forbidden for that position. In this sense, probing mainly removes false positives near the anchor boundary: it can rule out spuriously shallow assignments when a state was decoded into the near-anchor region but admits no compatible store match within the allowed hop budget. By contrast, direct anchors can correct both false positives and false negatives, since an exact match immediately fixes the depth.

This mechanism is most useful when $d_{\text{ref}}$ is kept modest for memory reasons. For example, under HTM, using $d_{\text{ref}} = 6$ is practical, whereas extending the exact store to $d_{\text{ref}} = 7$ or $8$ would require substantially more memory. Probe refinement provides a lightweight way to recover some of the benefits of a deeper store near this boundary, without incurring the full storage cost. After probing updates the feasible set, forward–backward inference is rerun under the revised constraints.

## C. Training Loss Definitions

This appendix gives the loss definitions used in Section 3.4. The model architecture and refinement–training loop are described in the main text.

**Model Outputs.** For a state $x$, the regression head predicts a scalar depth $\hat{d}(x)$, and the classification head defines

$$p_\theta(d \mid x) = \text{softmax}(\ell_\theta(x))_d, \qquad d \in \{0, \ldots, d_{\text{max}}\}.$$

**Objective.** The student is trained with

$$\mathcal{L} = \lambda_{\text{reg}}\mathcal{L}_{\text{MSE}} + \lambda_{\text{cls}}\mathcal{L}_{\text{CE}} + \lambda_{\text{nbr}}\mathcal{L}_{\text{nbr}} + \lambda_{\text{step}}\mathcal{L}_{\text{step}} + \lambda_{\text{glob}}\mathcal{L}_{\text{glob}} + \lambda_{\text{inv}}\mathcal{L}_{\text{inv}}.$$

**Supervised Losses.** Given decoded pseudo-labels $\hat{D}_b$, we train the regression and classification heads with

$$\mathcal{L}_{\text{MSE}} = \sum_b W_b(\hat{d}_b - \hat{D}_b)^2, \qquad \mathcal{L}_{\text{CE}} = \sum_b W_b \, \text{CE}(\ell_b, \hat{D}_b).$$

The mini-batch weights $W_b$ only rescale the supervised loss contributions of shallow exact labels and deeper pseudo-labels.

**Neighbor Consistency.** For trajectory position $(g, t)$, let

$$\delta_{g,t} = |\hat{d}_{g,t+1} - \hat{d}_{g,t}|.$$

The neighbor-consistency loss penalizes adjacent prediction jumps larger than margin $m$:

$$\mathcal{L}_{\text{nbr}} = \sum_{g,t} \mathbf{1}\{\delta_{g,t} > m\}\delta_{g,t}^2.$$

**Distributional Regularization.** Let $p_{g,t}(d) = p_\theta(d \mid X_{g,t})$. The step-prior term matches the average predicted distribution at each trajectory position to the fixed step-conditioned prior $\pi_t$:

$$\bar{p}_t(d) = \frac{1}{G}\sum_g p_{g,t}(d), \qquad \mathcal{L}_{\text{step}} = \frac{1}{S}\sum_t \text{KL}(\bar{p}_t \| \pi_t).$$

We also match the overall predicted depth marginal to the global prior $\rho(d) = S^{-1}\sum_t \pi_t(d)$:

$$\bar{p}(d) = \frac{1}{GS}\sum_{g,t} p_{g,t}(d), \qquad \mathcal{L}_{\text{glob}} = \text{KL}(\bar{p} \| \rho).$$

**Algorithm 2** Anchored HMM trajectory refinement

1: **Input:**
2:     store $\mathcal{S}$ $(d \leq d_{\text{ref}})$
3:     horizon $T = s_{\max}$, depth limit $d_{\max}$
4:     EMA teacher $f_{\bar{\theta}}$, probe scope $\Delta_{\text{probe}}$
5:     transition $A$, step priors $\pi_t$
6: **Output:** pseudo-label sequence $\hat{D}_{0:T}$
7: Retrieve $X_{0:d_{\text{ref}}}$ from $\mathcal{S}$ with anchors $D_t = t$
8: Roll out random moves to generate $X_{d_{\text{ref}}+1:T}$
9: **for** $t = d_{\text{ref}} + 1$ **to** $T$ **do**
10:     **if** $X_t \in \mathcal{S}$ at depth $d_m$ **then**
11:         $D_t := d_m$
12:     **else**
13:         $E_t(\cdot) \leftarrow f_{\bar{\theta}}(X_t)$
14:     **end if**
15: **end for**
16: Apply local step-feasibility masks
17: Run forward–backward with $(A, E)$ to obtain $\gamma_t(\cdot)$
18: Decode $\hat{D}_t := \arg\max_d \gamma_t(d)$
19:
20: **Probe refinement**
21: **for** each non-anchored $t$ with $\hat{D}_t \leq d_{\text{ref}} + \Delta_{\text{probe}}$ **do**
22:     Probe $X_t$ up to $\Delta_{\text{probe}}$ steps for a store match
23:     **if** match found at depth $d_m$ after $h$ hops **then**
24:         $D_t := d_m + h$
25:     **else**
26:         Mask depths $d \leq d_{\text{ref}} + \Delta_{\text{probe}}$
27:     **end if**
28: **end for**
29: If constraints changed, rerun forward–backward
30:
31: **Step-prior calibration**
32: Adjust $\gamma_t$ using step priors $\pi_t$ under anchor and mask constraints
33: Update $\hat{D}_t := \arg\max_d \gamma_t(d)$
34: **Return** $\hat{D}_{0:T}$

**Feasibility Penalty.** For trajectory state $X_{g,t}$, the constructed path gives the upper bound $D^\star(X_{g,t}) \leq t$. Let $h_{g,t} = 1$ indicate that $X_{g,t}$ is hard-anchored, either by construction in the trusted prefix or by a match to the frozen depth store, and let $a_{g,t}$ be its exact anchor depth. With $\bar{t} = \min(t, d_{\max})$, define

$$\mathcal{F}_{g,t} = \begin{cases} \{a_{g,t}\}, & h_{g,t} = 1, \\ \{d_{\text{ref}} + 1, \ldots, \bar{t}\}, & h_{g,t} = 0. \end{cases}$$

The auxiliary feasibility loss penalizes classification mass outside this set:

$$\mathcal{L}_{\text{inv}} = \sum_{g,t} W_{g,t}^{\text{eff}} \sum_{d \notin \mathcal{F}_{g,t}} p_\theta(d \mid X_{g,t}).$$

**Teacher Emissions.** For each non-fixed trajectory state $X_t$, the EMA teacher defines a temperature-scaled categorical likelihood $p_{\bar{\theta}}^{(\tau)}(d \mid X_t)$ and a regression-based Gaussian likelihood with mean $r_t$ and scale $\sigma_t$. The emission log-score is

$$e_t(d) = \eta_{\text{cls}} \log p_{\bar{\theta}}^{(\tau)}(d \mid X_t) - \eta_{\text{reg}} \frac{(d - r_t)^2}{2\sigma_t^2}.$$

The emissions are masked by anchors and feasibility constraints before forward–backward inference.

# D. CubeLocalRank Dataset and Evaluation Protocol

CubeLocalRank evaluates whether a learned depth predictor correctly ranks locally reachable states by true optimal depth on the $3 \times 3 \times 3$ Rubik's Cube under HTM.

### D.1. Reference States

We construct the reference set by starting from the solved cube and applying a uniformly random number of moves between 1 and 1000. This process produces approximately 170k samples. After deduplication, mainly removing repeated shallow configurations, the final dataset contains 167,536 unique reference states. We exclude the goal state, i.e., the solved cube, because all of its one-move neighbors have depth 1 and therefore do not define a meaningful ranking problem.

Each reference state $x$ is annotated with its exact HTM depth $D(x)$, computed using an optimal solver (Skalski).

### D.2. Local Neighborhoods

For each reference state $x$, we construct two local candidate sets.

**Level 1 (`lvl1`).** The `lvl1` neighborhood consists of the 18 one-move neighbors of $x$, obtained by applying each legal HTM move once. The associated evaluation problem is whether the model assigns the best score to at least one truly optimal immediate successor.

**Level 2 (`lvl2`).** The `lvl2` neighborhood is defined as a deduplicated two-step local neighborhood rather than the full set of $18 \times 18$ move sequences treated as distinct candidates. Immediate reversals and repeated states are removed, while the level-1 states are retained in the final candidate set. This yields 243 additional states beyond `lvl1`, for a total of 261 candidates per reference state under `lvl2` evaluation.

All candidate states are also annotated with exact HTM depths using the same solver. Thus, CubeLocalRank provides two local-ranking settings:

- `lvl1`: ranking among 18 one-move candidates;

- `lvl2`: ranking among 261 candidates in the deduplicated two-step local neighborhood.

Figure 6 illustrates the neighborhood structure used in CubeLocalRank.

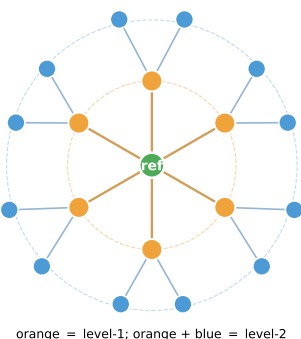

orange = level-1; orange + blue = level-2

*Figure 6.* **CubeLocalRank neighborhood structure.** Level 1 consists of the 18 one-move neighbors of a reference state. Level 2 extends this to a deduplicated two-step local neighborhood while retaining the level-1 states.

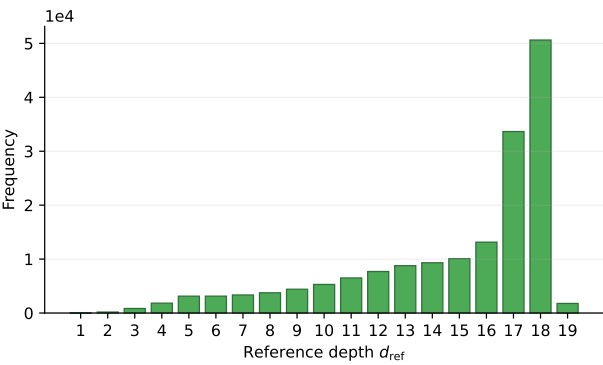

*Figure 7.* **Frequency of reference-state optimal depths in Cube-LocalRank.** Distribution of exact reference depths $d_{\text{ref}}$ over the 167,536 unique reference states in the benchmark.

### D.3. Depth Distribution of Reference States

Figure 7 shows the distribution of reference-state optimal depths in CubeLocalRank. The dataset contains no reference states with optimal depth 20. This is consistent with prior estimates indicating that depth-20 states are extraordinarily rare under random sampling, on the order of approximately one configuration in every 90 billion random scrambles (Kociemba, 2014). Consistent with this rarity, we also do not observe any depth-20 states within the corresponding level-1 or level-2 neighborhoods induced by our reference set.

### D.4. Compact Representation

To keep the released benchmark compact, we store each reference configuration together with the move sequence used to generate it, along with the action sequences defining its level-1 and level-2 neighborhoods, rather than materializing every candidate state explicitly. This compressed representation occupies approximately 12MB, whereas the fully expanded `lvl1` and `lvl2` neighborhoods would require approximately 1.4GB and 18GB, respectively.

### D.5. Evaluation Metric

CubeLocalRank evaluates whether a learned depth predictor correctly identifies the best local candidate within a fixed neighborhood. For each reference state $x$ with true depth $D(x)$, let $\mathcal{N}(x)$ denote its candidate set. In `lvl1`, this set contains the 18 one-move neighbors. In `lvl2`, it contains the 261-state deduplicated two-step local neighborhood.

Each candidate $y \in \mathcal{N}(x)$ has exact true depth $D(y)$, while the model assigns a predicted depth score $\hat{D}(y)$, where lower is better. The evaluation checks whether the model assigns the top rank to at least one truly optimal candidate.

The set of truly best candidates is

$$\mathcal{B}(x) = \arg \min_{y \in \mathcal{N}(x)} D(y),$$

and the set of predicted-best candidates is

$$\hat{\mathcal{B}}(x) = \arg \min_{y \in \mathcal{N}(x)} \hat{D}(y).$$

The evaluation is tie-aware: if multiple candidates receive the same minimum predicted score, they are all treated as predicted-best candidates. A reference state is counted as correct if

$$\mathcal{B}(x) \cap \hat{\mathcal{B}}(x) \neq \emptyset.$$

The Rank at reference depth $d$ is therefore

$$\text{Rank}(d) = \frac{\sum_{x:D(x)=d} \mathbf{1}\Big[\mathcal{B}(x) \cap \hat{\mathcal{B}}(x) \neq \emptyset\Big]}{|\{x : D(x) = d\}|}.$$

This measures, among all reference states at depth $d$, how often the model places a truly optimal local candidate at the top rank.

We also report a cumulative aggregate score up to depth $d$. Let $C(j)$ denote the number of correct local-ranking decisions at reference depth $j$, and let $N(j)$ denote the number of reference states at depth $j$. The Aggregate Rank up to depth $d$ is

$$\text{Agg. Rank}(\leq d) = \frac{\sum_{j \leq d} C(j)}{\sum_{j \leq d} N(j)} = \frac{\sum_{j \leq d} N(j)\,\text{Rank}(j)}{\sum_{j \leq d} N(j)}.$$

Unlike an unweighted average across depths, this aggregate is weighted by the number of reference states at each depth. It therefore measures how often, over all reference states with depth at most $d$, the model places a truly optimal local candidate at the top rank.

## E. Computational Budget and Cost Analysis

This appendix reports additional measurements on training cost and search-time comparability for the $3\times3\times3$ Rubik's Cube in QTM under the standard DeepCubeA evaluation protocol.

**Hardware.** All runtime measurements were obtained on a single NVIDIA RTX A6000 GPU (48 GB memory).

### E.1. Training Cost of Anchored HMM Refinement

The anchored HMM pipeline adds a one-time preprocessing cost for constructing the exact-depth store and a small per-iteration overhead.

**Exact-Depth Store.** The store is built once by breadth-first search up to $d \leq d_{\mathrm{ref}}$. For the $3{\times}3{\times}3$ cube in QTM, representative sizes are: depth 4: $\sim$10k, depth 5: $\sim$94k, depth 6: $\sim$0.88M, and depth 7: $\sim$8.2M states. In practice, $d_{\mathrm{ref}} = 7$ requires about 37 s of preprocessing and 721 MB of memory. This is a one-time cost and is not part of the per-iteration training budget. The matching depth $d_{\mathrm{ref}}$ can also be chosen separately for training and search, and the training-side value may vary over training.

**Per-Iteration Cost.** We measure iteration cost with approximately 1M training samples per iteration. For a CayleyPy-style baseline, the combined data-generation and training time is 3.11 s, including 2.14 s for the training step. For our method, total cost is 3.42 s at $d_{\mathrm{ref}} = 1$ and 3.18 s at $d_{\mathrm{ref}} = 7$, with training time $\approx 2.30$ s at $d_{\mathrm{ref}} = 7$. Thus, refinement makes the training step slightly more expensive, but total iteration cost remains comparable; at larger $d_{\mathrm{ref}}$, it can even decrease because more states are resolved directly by anchors. At practical settings, the overhead is small (about 2–3% at $d_{\mathrm{ref}} = 7$).

**Effect of Probing.** At $d_{\mathrm{ref}} = 7$, bounded probing increases total cost to $\sim$3.24 s for probe = 1 ($\sim$4%) and $\sim$3.51 s for probe = 2 ($\sim$13%).

**Numerical Precision during Training.** Training, including EMA teacher updates, is performed in FP32, which we found more stable for continuously updated parameters and refinement-dependent supervision.

### E.2. Training Schedule and Checkpoint Selection

Training uses online trajectory generation with an SGDR-style restart schedule. The initial cycle length is $C_0 = 200$ epochs, and each subsequent cycle is multiplied by 1.5. Each epoch generates 40,000 fresh trajectories, which are optimized in mini-batches of size 2,000.

Reported checkpoints are taken at restart boundaries, except for the $3{\times}3{\times}3$ QTM 52B run, which was stopped at the target sample budget before the current cycle ended.

### E.3. Search Budget Comparability

All beam-search comparisons in the main paper use matched search settings: the same beam size, expansion rules, and stagnation handling are used across methods. To isolate the role of the learned heuristic, we also evaluate our method with $d_{\mathrm{ref}} = 0$, which recovers a CayleyPy-style search-time setup.

At beam size $2^{18}$ on the 1k DeepCubeA benchmark, search effort is nearly identical:

- **CayleyPy-style (our run):** 12.2 s, 4.24M expansions

- **Ours** ($d_{\mathrm{ref}} = 0$)**:** 12.1 s, 4.21M expansions

Thus, the baseline and our implementation use essentially the same search budget under matched settings.

We also evaluated the effect of search-time matching depth on the $3{\times}3{\times}3$ cube in QTM. Over the beam range used in our main comparisons ($2^{12}$–$2^{24}$), varying $d_{\mathrm{ref}}$ during search had no effect on solution quality. This is consistent with the learned heuristic already modeling shallow depths accurately, so explicit matching in this regime improves efficiency rather than solvability. For example, at beam $2^{12}$, CayleyPy solves 26.1% of instances optimally, whereas our model solves 35.9%; this advantage is preserved even when $d_{\mathrm{ref}} = 0$ at search time. These results indicate that the quality gains reported in the main paper arise from the learned heuristic rather than from search-time matching.

When intermediate matching is enabled ($d_{\mathrm{ref}} > 0$), the main effect is reduced runtime and fewer redundant expansions under the same beam constraint. At beam size $2^{18}$, runtime decreases from 12.1 s to 7.14 s per instance. At beam $2^{24}$ with $d_{\mathrm{ref}} = 7$, our search is more than 200 s faster on average per instance than CayleyPy. The same pattern appears under batch A*:

- **DeepCubeA:** 75.6 s, 8.2M expansions

- **Ours:** 45.4 s, 5.4M expansions

**Numerical Precision during Search.** Search uses FP16 model evaluation, following CayleyPy. This substantially reduces runtime and is used for the reported speed comparisons. At selected beam sizes, FP32 improved solution quality only marginally (at most one additional solved instance on the 1k DeepCubeA benchmark, about 0.1%) while roughly doubling runtime.

## F. Additional Large-Puzzle Experiments and Diagnostics

This section provides supporting diagnostics for the large-puzzle experiments, where exact optimal depths are un-

*Table 3*. $4\times4\times4$ **assumed depth-cap sensitivity.** Solvability is measured on 43 Santa Claus 2023 instances using beam $2^{16}$.

| Assumed depth cap | Solved at beam $2^{16}$ |
|---|---|
| 20 | 22/43 (51.2%) |
| 25 | 29/43 (67.4%) |
| 35 | 36/43 (83.7%) |
| 45 | 35/43 (81.4%) |
| 55 | 38/43 (88.4%) |

available and evaluation sets are small. We report assumed depth-cap sensitivity, fixed-beam solution-length comparisons, and a limited $6\times6\times6$ study.

### F.1. Assumed Depth Caps on Larger Cubes

For larger cubes, we use an *assumed depth cap* only to define the finite output space and prior-based objectives; it should not be interpreted as a proven diameter.

For $4\times4\times4$, we tested several assumed depth caps and evaluated the resulting models at beam $2^{16}$ on the 43 Santa Claus 2023 instances. The results are shown in Table 3. Small caps under-cover the effective search range and reduce solvability. Larger caps are more stable, although the trend is not perfectly monotone on this small evaluation set. We use depth cap 45 in the reported $4\times4\times4$ experiments to match the setting used by CayleyPy, rather than tuning the cap to maximize the diagnostic below.

For $5\times5\times5$, we use depth cap 50, while CayleyPy uses 65. We did not increase the cap further in the reported study because the model already reaches full solvability within the evaluated beam range, and increasing the cap enlarges the output space and training cost. Thus, the $5\times5\times5$ cap should be interpreted as a practical modeling choice under a fixed compute budget, not as a claim about the true diameter of the puzzle.

### F.2. Single-Beam Solution Length and Search-Budget Comparison

Solvability measures whether a method finds a valid solution, but it does not capture the length of the returned path. We therefore also compare mean solution length for settings where all instances are solved.

For $4\times4\times4$, our 8B single model solves all 43 Santa Claus 2023 instances at beams $2^{18}$, $2^{19}$, $2^{20}$, and $2^{21}$. As shown in Table 4, increasing the beam improves average solution length. At beam $2^{21}$, our single model obtains mean length 48.70. For comparison, the CayleyPy 8B single model obtains mean length about 49.0 at beam $2^{24}$, and the CayleyPy 29-model ensemble obtains mean length about 46.5 at the same beam. Thus, our single model reaches comparable single-model solution quality using a beam budget that is $8\times$ smaller than $2^{24}$. Since runtime scales approximately

*Table 4*. $4\times4\times4$ **solution length at fixed beams.** Our 8B model solves all 43 Santa Claus 2023 instances at each listed beam. CayleyPy values are included for context at their larger reported beam.

| Method | Beam | Mean length |
|---|---|---|
| Ours, single model, 8B | $2^{18}$ | 52.28 |
| Ours, single model, 8B | $2^{19}$ | 51.26 |
| Ours, single model, 8B | $2^{20}$ | 49.77 |
| Ours, single model, 8B | $2^{21}$ | 48.70 |
| CayleyPy, single model, 8B | $2^{24}$ | 49.0 |
| CayleyPy, 29-model ensemble | $2^{24}$ | 46.5 |

linearly with beam size in our experiments, this corresponds to a substantially smaller search budget under the same single-model parameter setting. The CayleyPy ensemble obtains shorter paths, but uses both a much larger beam and multiple models.

For $5\times5\times5$, our 27B single model solves all 19 Santa Claus 2023 instances at beam $2^{18}$, with mean solution length 137.7. We do not report a corresponding CayleyPy mean length over the full test set, because the released single model does not solve all instances within the beam range used in our solvability evaluation. Reporting an average only over the solved subset would not be comparable to an average over all instances, so we omit this comparison.

### F.3. Limited $6\times6\times6$ Study

We also ran a limited $6\times6\times6$ experiment. This setting is substantially more expensive, and the Santa Claus 2023 benchmark contains only six $6\times6\times6$ instances. In a short training run, the model solved one of the six instances when searched up to beam $2^{18}$. We did not continue this experiment further because the combination of high search cost and a six-instance test set makes it difficult to draw a reliable scaling conclusion. Accordingly, we treat $6\times6\times6$ as a compute-limited preliminary setting and do not use it as a main result.

### F.4. Fixed-Budget Interpretation

The large-puzzle experiments should be interpreted as fixed-budget comparisons, not exhaustive scaling studies. As puzzle size increases, the effective depth range, training cost, and downstream beam-search cost all grow, while the available evaluation sets remain small. We therefore report whether a valid solution is found within the specified search budget, and do not claim optimality on these larger domains.

## G. Extended Results

This section presents extended quantitative results supporting the main paper. We report detailed downstream beam-search performance on DeepCubeA and search-free heuristic quality measured by CubeLocalRank across a range of

*Table 5.* **HTM beam-search results.** Optimality and mean solution length on the DeepCubeA benchmark as functions of training size and beam.

**(a) Optimality.**

| Train ($\times 10^9$) | $2^{12}$ | $2^{13}$ | $2^{14}$ | $2^{15}$ | $2^{16}$ | $2^{17}$ | $2^{18}$ |
|---|---|---|---|---|---|---|---|
| 0.25 | 0.075 | 0.096 | 0.128 | 0.173 | 0.217 | 0.273 | 0.352 |
| 0.5 | 0.097 | 0.136 | 0.175 | 0.230 | 0.276 | 0.354 | 0.430 |
| 1 | 0.125 | 0.162 | 0.215 | 0.262 | 0.340 | 0.418 | 0.495 |
| 1.6 | 0.136 | 0.182 | 0.234 | 0.286 | 0.363 | 0.446 | 0.534 |
| 2.7 | 0.146 | 0.194 | 0.235 | 0.302 | 0.381 | 0.484 | 0.568 |
| 4.2 | 0.162 | 0.202 | 0.263 | 0.321 | 0.412 | 0.498 | 0.599 |
| 6.4 | 0.174 | 0.208 | 0.266 | 0.351 | 0.430 | 0.525 | 0.613 |
| 10 | 0.178 | 0.222 | 0.280 | 0.360 | 0.455 | 0.531 | 0.633 |
| 15 | 0.185 | 0.234 | 0.298 | 0.372 | 0.465 | 0.555 | 0.641 |
| 22 | 0.190 | 0.243 | 0.304 | 0.397 | 0.484 | 0.569 | 0.666 |
| 34 | 0.194 | 0.251 | 0.312 | 0.401 | 0.496 | 0.580 | 0.667 |
| 52 | 0.202 | 0.252 | 0.320 | 0.410 | 0.505 | 0.597 | 0.681 |

**(b) Mean solution length.**

| Train ($\times 10^9$) | $2^{12}$ | $2^{13}$ | $2^{14}$ | $2^{15}$ | $2^{16}$ | $2^{17}$ | $2^{18}$ |
|---|---|---|---|---|---|---|---|
| 0.25 | 20.35 | 19.89 | 19.56 | 19.24 | 19.00 | 18.78 | 18.58 |
| 0.5 | 19.92 | 19.57 | 19.25 | 18.98 | 18.79 | 18.57 | 18.41 |
| 1 | 19.65 | 19.36 | 19.07 | 18.84 | 18.61 | 18.43 | 18.30 |
| 1.6 | 19.56 | 19.24 | 18.98 | 18.75 | 18.54 | 18.38 | 18.24 |
| 2.7 | 19.46 | 19.16 | 18.93 | 18.71 | 18.52 | 18.33 | 18.20 |
| 4.2 | 19.40 | 19.09 | 18.85 | 18.64 | 18.45 | 18.30 | 18.16 |
| 6.4 | 19.33 | 19.05 | 18.81 | 18.58 | 18.42 | 18.26 | 18.13 |
| 10 | 19.25 | 18.99 | 18.76 | 18.56 | 18.38 | 18.24 | 18.10 |
| 15 | 19.21 | 18.95 | 18.73 | 18.52 | 18.35 | 18.21 | 18.09 |
| 22 | 19.18 | 18.89 | 18.69 | 18.48 | 18.32 | 18.20 | 18.06 |
| 34 | 19.16 | 18.87 | 18.66 | 18.46 | 18.30 | 18.17 | 18.05 |
| 52 | 19.11 | 18.85 | 18.62 | 18.43 | 18.28 | 18.15 | 18.04 |

*Table 6.* **QTM beam-search results.** Optimality and mean solution length on the DeepCubeA benchmark as functions of beam width and training size.

| Beam | 0.27B | | 1B | | 10B | | 52B | |
|---|---|---|---|---|---|---|---|---|
| | Opt. | Len. | Opt. | Len. | Opt. | Len. | Opt. | Len. |
| $2^{12}$ | 0.141 | 23.45 | 0.205 | 22.83 | 0.317 | 22.26 | 0.359 | 22.11 |
| $2^{13}$ | 0.186 | 22.95 | 0.278 | 22.42 | 0.402 | 21.98 | 0.448 | 21.85 |
| $2^{14}$ | 0.243 | 22.60 | 0.356 | 22.12 | 0.493 | 21.75 | 0.534 | 21.62 |
| $2^{15}$ | 0.326 | 22.24 | 0.435 | 21.88 | 0.564 | 21.54 | 0.609 | 21.44 |
| $2^{16}$ | 0.398 | 21.99 | 0.532 | 21.64 | 0.645 | 21.36 | 0.688 | 21.27 |
| $2^{17}$ | 0.481 | 21.77 | 0.603 | 21.46 | 0.713 | 21.22 | 0.746 | 21.15 |
| $2^{18}$ | 0.569 | 21.55 | 0.667 | 21.32 | 0.778 | 21.09 | 0.805 | 21.03 |
| $2^{19}$ | 0.646 | 21.37 | 0.731 | 21.18 | 0.828 | 20.98 | 0.855 | 20.93 |
| $2^{20}$ | 0.705 | 21.24 | 0.789 | 21.06 | 0.875 | 20.89 | 0.901 | 20.84 |
| $2^{21}$ | 0.756 | 21.13 | 0.835 | 20.97 | 0.901 | 20.84 | 0.921 | 20.80 |
| $2^{22}$ | 0.813 | 21.01 | 0.883 | 20.87 | 0.925 | 20.79 | 0.944 | 20.75 |
| $2^{23}$ | 0.863 | 20.91 | 0.912 | 20.81 | 0.946 | 20.75 | 0.963 | 20.71 |
| $2^{24}$ | 0.898 | 20.84 | 0.938 | 20.76 | 0.970 | 20.70 | 0.977 | 20.68 |

### G.2. Additional CubeLocalRank Results

We provide additional search-free heuristic evaluation results using CubeLocalRank, extending the analysis in Section 4.1. Results are reported for both level-1 and level-2 neighborhoods, stratified by reference depth and training set size.

**Level-1 Neighborhoods.** Table 7 reports the corresponding results for level-1 neighborhoods.

**Level-2 Neighborhoods.** Table 8 reports the corresponding results for level-2 neighborhoods.

### G.3. Local Ranking Quality and Beam Search Performance

We examine how search-free heuristic quality relates to downstream search performance. Figure 8 compares aggregate CubeLocalRank Rank@1 accuracy with DeepCubeA beam-search accuracy for the same set of models. Under HTM, improvements in local ranking quality (Figure 8a) are accompanied by higher beam-search accuracy (Figure 8b), especially at small to moderate beam widths, where search behavior is most strongly influenced by heuristic guidance.

At larger beam widths, beam-search performance begins to saturate and the gap between models narrows. This is consistent with the expected transition from strongly heuristic-guided search toward broader exploration as the beam budget increases. The same qualitative trend is observed under QTM (Figure 8c), where stronger models again provide the clearest gains at practical beam sizes, while differences diminish as the search budget becomes very large.

training set sizes and search budgets. These results document how performance scales with data and beam width and enable direct comparison between local heuristic quality and downstream search behavior under controlled settings.

### G.1. Downstream Beam Search Performance on DeepCubeA

For HTM experiments, models are trained with reference depth $d_{\text{ref}} = 6$ and probe radius $p = 2$; QTM experiments use $d_{\text{ref}} = 7$ and $p = 2$. We evaluate multiple training set sizes, including configurations chosen to match prior work and additional intermediate scales to illustrate performance trends.

**Half-Turn Metric (HTM).** Table 5 reports beam-search optimality and mean solution length on the DeepCubeA benchmark under HTM as functions of training size and beam width.

**Quarter-Turn Metric (QTM).** Table 6 reports beam-search optimality and mean solution length on the Deep-CubeA benchmark under QTM as functions of beam width and training size.

## H. Additional Ablation Studies

This section provides supporting diagnostics for Section 5. Unless stated otherwise, all values are CubeLocalRank Level-1 Aggregate Rank@1 on the $3 \times 3 \times 3$ Rubik's Cube

*Table 7.* **CubeLocalRank level-1 results on the** $3 \times 3 \times 3$ **Rubik's Cube under HTM.** Per-depth Rank@1 and overall Aggregate Rank@1 are reported as a function of training size.

| Training size ($\times 10^9$) samples | 1 | 2 | 3 | 4 | 5 | 6 | 7 | 8 | 9 | 10 | 11 | 12 | 13 | 14 | 15 | 16 | 17 | 18 | 19 | Overall Aggregate Rank@1 |
|---|---|---|---|---|---|---|---|---|---|---|---|---|---|---|---|---|---|---|---|---|
| 0.25 | 1 | 1 | 1 | 1 | 1 | 0.94 | 0.89 | 0.79 | 0.67 | 0.54 | 0.43 | 0.34 | 0.27 | 0.23 | 0.19 | 0.17 | 0.17 | 0.37 | 0.93 | 0.36 |
| 0.5 | 1 | 1 | 1 | 1 | 1 | 0.97 | 0.93 | 0.85 | 0.74 | 0.6 | 0.49 | 0.38 | 0.3 | 0.25 | 0.2 | 0.18 | 0.18 | 0.39 | 0.94 | 0.39 |
| 1 | 1 | 1 | 1 | 1 | 1 | 0.98 | 0.96 | 0.89 | 0.77 | 0.64 | 0.52 | 0.41 | 0.32 | 0.26 | 0.21 | 0.18 | 0.19 | 0.4 | 0.95 | 0.4 |
| 1.6 | 1 | 1 | 1 | 1 | 1 | 0.99 | 0.97 | 0.91 | 0.8 | 0.67 | 0.54 | 0.43 | 0.33 | 0.27 | 0.22 | 0.19 | 0.19 | 0.4 | 0.95 | 0.41 |
| 2.7 | 1 | 1 | 1 | 1 | 1 | 1 | 0.98 | 0.93 | 0.82 | 0.69 | 0.56 | 0.44 | 0.34 | 0.28 | 0.22 | 0.19 | 0.19 | 0.4 | 0.96 | 0.41 |
| 4.2 | 1 | 1 | 1 | 1 | 1 | 1 | 0.99 | 0.94 | 0.84 | 0.71 | 0.58 | 0.45 | 0.34 | 0.28 | 0.23 | 0.19 | 0.19 | 0.4 | 0.96 | 0.42 |
| 6.4 | 1 | 1 | 1 | 1 | 1 | 1 | 0.99 | 0.95 | 0.85 | 0.72 | 0.59 | 0.46 | 0.35 | 0.28 | 0.23 | 0.2 | 0.19 | 0.4 | 0.95 | 0.42 |
| 10 | 1 | 1 | 1 | 1 | 1 | 1 | 1 | 0.96 | 0.86 | 0.74 | 0.6 | 0.47 | 0.36 | 0.29 | 0.23 | 0.2 | 0.19 | 0.4 | 0.96 | 0.42 |
| 15 | 1 | 1 | 1 | 1 | 1 | 1 | 1 | 0.97 | 0.87 | 0.74 | 0.62 | 0.48 | 0.36 | 0.29 | 0.23 | 0.2 | 0.19 | 0.4 | 0.95 | 0.42 |
| 22 | 1 | 1 | 1 | 1 | 1 | 1 | 1 | 0.97 | 0.87 | 0.76 | 0.61 | 0.48 | 0.37 | 0.29 | 0.24 | 0.2 | 0.19 | 0.4 | 0.96 | 0.43 |
| 34 | 1 | 1 | 1 | 1 | 1 | 1 | 1 | 0.98 | 0.89 | 0.77 | 0.62 | 0.49 | 0.38 | 0.3 | 0.24 | 0.2 | 0.2 | 0.4 | 0.95 | 0.43 |
| 52 | 1 | 1 | 1 | 1 | 1 | 1 | 1 | 0.98 | 0.89 | 0.77 | 0.63 | 0.5 | 0.37 | 0.3 | 0.24 | 0.2 | 0.2 | 0.4 | 0.95 | 0.43 |

*Table 8.* **CubeLocalRank level-2 results on the** $3 \times 3 \times 3$ **Rubik's Cube under HTM.** Per-depth Rank@1 and overall Aggregate Rank@1 are reported as a function of training size.

| Training size ($\times 10^9$) samples | 1 | 2 | 3 | 4 | 5 | 6 | 7 | 8 | 9 | 10 | 11 | 12 | 13 | 14 | 15 | 16 | 17 | 18 | 19 | Overall Aggregate Rank@1 |
|---|---|---|---|---|---|---|---|---|---|---|---|---|---|---|---|---|---|---|---|---|
| 0.25 | 1 | 1 | 1 | 1 | 1 | 0.99 | 0.91 | 0.82 | 0.65 | 0.47 | 0.31 | 0.2 | 0.13 | 0.094 | 0.06 | 0.046 | 0.04 | 0.08 | 0.34 | 0.2 |
| 0.5 | 1 | 1 | 1 | 1 | 1 | 1 | 0.95 | 0.89 | 0.73 | 0.55 | 0.38 | 0.26 | 0.16 | 0.11 | 0.072 | 0.054 | 0.044 | 0.091 | 0.38 | 0.22 |
| 1 | 1 | 1 | 1 | 1 | 1 | 1 | 0.98 | 0.93 | 0.78 | 0.62 | 0.43 | 0.28 | 0.18 | 0.12 | 0.074 | 0.058 | 0.048 | 0.097 | 0.4 | 0.23 |
| 1.6 | 1 | 1 | 1 | 1 | 1 | 1 | 0.99 | 0.95 | 0.82 | 0.66 | 0.47 | 0.31 | 0.2 | 0.13 | 0.078 | 0.063 | 0.05 | 0.098 | 0.39 | 0.24 |
| 2.7 | 1 | 1 | 1 | 1 | 1 | 1 | 0.99 | 0.97 | 0.85 | 0.68 | 0.5 | 0.32 | 0.21 | 0.13 | 0.083 | 0.063 | 0.05 | 0.1 | 0.41 | 0.25 |
| 4.2 | 1 | 1 | 1 | 1 | 1 | 1 | 1 | 0.98 | 0.88 | 0.71 | 0.52 | 0.34 | 0.21 | 0.14 | 0.085 | 0.064 | 0.052 | 0.1 | 0.39 | 0.25 |
| 6.4 | 1 | 1 | 1 | 1 | 1 | 1 | 1 | 0.99 | 0.9 | 0.74 | 0.54 | 0.35 | 0.22 | 0.15 | 0.087 | 0.065 | 0.053 | 0.1 | 0.41 | 0.26 |
| 10 | 1 | 1 | 1 | 1 | 1 | 1 | 1 | 0.99 | 0.92 | 0.76 | 0.55 | 0.36 | 0.23 | 0.15 | 0.092 | 0.064 | 0.052 | 0.1 | 0.42 | 0.26 |
| 15 | 1 | 1 | 1 | 1 | 1 | 1 | 1 | 0.99 | 0.93 | 0.78 | 0.57 | 0.38 | 0.24 | 0.16 | 0.094 | 0.066 | 0.053 | 0.1 | 0.4 | 0.26 |
| 22 | 1 | 1 | 1 | 1 | 1 | 1 | 1 | 1 | 0.94 | 0.79 | 0.58 | 0.39 | 0.25 | 0.16 | 0.097 | 0.066 | 0.054 | 0.1 | 0.4 | 0.27 |
| 34 | 1 | 1 | 1 | 1 | 1 | 1 | 1 | 1 | 0.95 | 0.8 | 0.6 | 0.4 | 0.26 | 0.17 | 0.1 | 0.068 | 0.054 | 0.1 | 0.41 | 0.27 |
| 52 | 1 | 1 | 1 | 1 | 1 | 1 | 1 | 1 | 0.96 | 0.81 | 0.61 | 0.41 | 0.26 | 0.17 | 0.11 | 0.067 | 0.054 | 0.1 | 0.42 | 0.27 |

under HTM. Aggregate Rank@1 is reported on a 0–1 scale; a difference of 0.002 corresponds to 0.2 percentage points. Because some diagnostics use different matched settings, comparisons should be made within each table or figure panel. All reported scalar metrics are rounded to three significant figures.

### H.1. Trajectory-Level Refinement Baselines

The first block of Table 2 compares the full anchored HMM against simpler pseudo-labeling alternatives. The baselines differ in whether they use trajectory-level inference.

The *prior + anchors, uniform emissions* baseline still runs forward–backward inference. It retains the transition prior and exact anchors, but replaces learned emissions at non-anchored positions with a uniform distribution. This tests whether the prior and anchors alone are sufficient once HMM smoothing is available.

The *prior-sampled pseudo-labeling, no HMM* baseline removes forward–backward inference. Labels are instead sampled independently from the step-conditioned prior $\pi_t(d)$ at each trajectory position. This tests whether the prior alone provides useful supervision without trajectory-level smoothing.

These comparisons are distinct from the reference-depth sensitivity study in Section H.3. In particular, $d_{\mathrm{ref}} = 0$ is not a clean "no-anchor" HMM ablation: with only the solved state in the exact region, all legal moves from the known state increase depth, so the induced depth-change prior is also poorly grounded. We therefore interpret $d_{\mathrm{ref}} = 0$ as a joint sensitivity test of reference-depth coverage and prior construction, not as an isolated anchor ablation.

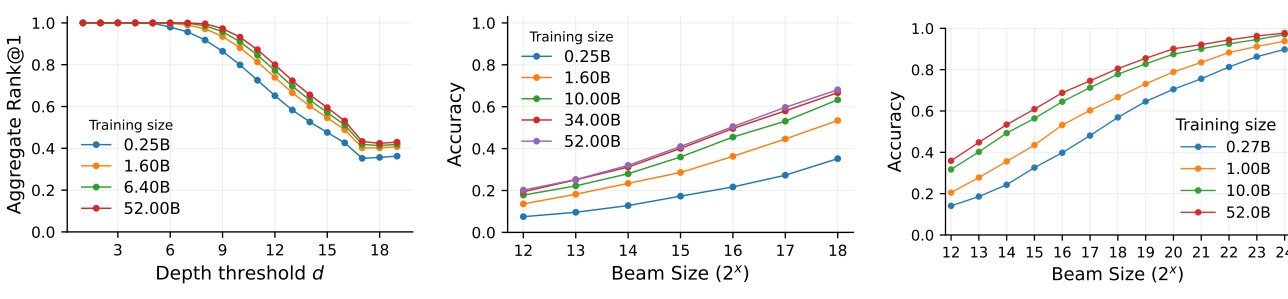

*(a)* CubeLocalRank aggregate Rank@1 (HTM, level-1).

*(b)* Beam-search accuracy on DeepCubeA (HTM).

*(c)* Beam-search accuracy on DeepCubeA (QTM).

*Figure 8.* **From local ranking quality to downstream search performance.** (a) Aggregate Rank@1 accuracy on CubeLocalRank under HTM as a function of training set size. (b) Beam-search accuracy on DeepCubeA under HTM for the same set of models, evaluated across beam budgets. (c) Beam-search accuracy on DeepCubeA under QTM across beam budgets. Across both metrics, stronger local ranking quality is associated with improved search performance, with the largest gains appearing at practical beam sizes.

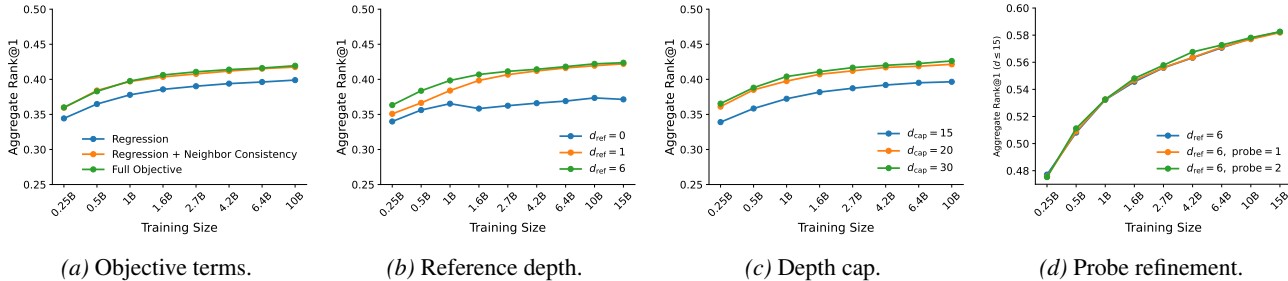

*(a)* Objective terms.

*(b)* Reference depth.

*(c)* Depth cap.

*(d)* Probe refinement.

*Figure 9.* **Additional ablation diagnostics.** (a) Objective ablation across training scale. Regression is the base supervised loss; neighbor consistency gives the main additional improvement, while the full objective remains close in local ranking and supplies distributional predictions for HMM emissions. (b) Reference-depth sensitivity. Shallow exact anchors are important, but deeper stores provide diminishing returns as training scale increases. (c) Provisional depth-cap sensitivity. A cap that is too small undercovers the useful training range, while larger caps behave similarly. (d) Probe refinement. Bounded probing improves intermediate-depth ranking, especially at medium training scales.

## H.2. Objective Terms

Scalar depth regression is the base supervised loss in all objective variants, because the regression head is the scalar heuristic used for ranking and search. The objective ablations ask which additional structural or distributional terms improve this baseline.

Figure 9a compares regression-only training, regression plus neighbor consistency, and the full objective across training scale. The corresponding values are shown in Table 9. Here, REG denotes scalar depth regression and NBR denotes the neighbor-consistency loss $\mathcal{L}_{\text{nbr}}$.

At 10B samples, adding neighbor consistency improves regression-only training by 1.86 percentage points using the unrounded values. Adding the remaining classification and KL terms gives only a small additional Rank@1 improvement beyond REG+NBR. Thus, neighbor consistency is the main structural regularizer for local ranking, while the distributional terms mainly support stable emissions for HMM refinement.

Table 10 reports a separate 200M diagnostic that isolates

objective components. Rows are cumulative until $\text{KL}_{\text{glob}}$; the final rows add NBR and vary its margin. The classification and KL terms improve the depth-distribution head used for emissions while leaving regression-head Rank@1 nearly unchanged. Adding NBR gives the main local-ranking gain, and performance is weakly sensitive to the margin once the loss is enabled.

Table 11 evaluates emission construction for HMM refinement. For classification-based emissions, temperature rescales the classification logits before forming the categorical likelihood. Rank@1 changes little across a broad temperature range, suggesting that the classification emission is not sensitive to this calibration. We also report the selected regression likelihood and the combined regression–classification emission used in the final model. Because we did not exhaustively sweep regression-likelihood parameters, this table should be interpreted as an emission-construction diagnostic rather than a full calibration study.

*Table 9.* **Objective ablation across training scale.** Values are Aggregate Rank@1. REG denotes scalar depth regression, NBR denotes neighbor consistency, and Full includes the auxiliary classification and KL terms.

| Train | REG | REG + NBR | Full |
|---|---|---|---|
| 0.25B | 0.338 | 0.360 | 0.360 |
| 0.5B | 0.365 | 0.384 | 0.383 |
| 1B | 0.378 | 0.397 | 0.398 |
| 1.6B | 0.386 | 0.404 | 0.406 |
| 2.7B | 0.390 | 0.408 | 0.411 |
| 4.2B | 0.394 | 0.412 | 0.414 |
| 6.4B | 0.396 | 0.415 | 0.416 |
| 10B | 0.399 | 0.418 | 0.419 |

*Table 10.* **Objective components at 200M samples.** Rows are cumulative until $KL_{glob}$. The final three rows add NBR to the same objective and vary its margin. Rank@1 is measured from the regression head; classification accuracy is measured on the depth-classification task.

| Objective | Margin | Rank@1 | Cls. acc. |
|---|---|---|---|
| REG | – | 0.327 | – |
| + CLS | – | 0.325 | 0.440 |
| + $KL_{step}$ | – | 0.326 | 0.447 |
| + $KL_{glob}$ | – | 0.326 | 0.450 |
| + NBR | 1.1 | **0.344** | **0.454** |
| + NBR | 1.2 | 0.343 | 0.453 |
| + NBR | 1.3 | 0.342 | 0.453 |

## H.3. Reference Depth and Depth Cap

Figure 9b shows the effect of the exact-depth reference store. Moving from $d_{ref} = 0$ to 1 gives a clear gain, while $d_{ref} = 1$ approaches $d_{ref} = 6$ as training scale grows. A representative 1B diagnostic is shown in Table 12.

The $d_{ref} = 0$ setting should not be interpreted as a clean HMM without anchors. With only the solved state in the exact region, the empirical depth-change statistics are degenerate: every legal move from depth 0 increases depth. Extending this behavior beyond the solved state produces a poorly grounded prior. This ablation therefore shows that nontrivial exact-depth coverage is needed both for anchoring and for reliable prior construction.

Figure 9c compares provisional depth caps. A cap of 15 undercovers the effective training range and degrades performance. Caps of 20 and 30 behave similarly, so for the $3 \times 3 \times 3$ cube under HTM we use $d_{cap} = 20$, matching the known HTM diameter while avoiding unnecessary unsupported tail classes.

## H.4. Probe Refinement

Probe refinement extends the effect of the exact-depth store without explicitly storing deeper BFS layers. Its effect is most visible at intermediate depths, so Figure 9d reports Aggregate Rank@1 restricted to $d \leq 15$. Probing improves sample efficiency at medium training scales, while the gap

*Table 11.* **Emission-construction diagnostic at 200M samples.** Values are Rank@1 / classification accuracy. CLS temperature rescales classification logits before emission construction. The REG likelihood uses the selected discretized-Gaussian calibration (scale = 0.5, floor = 0.5).

| Emission setting | Rank@1 | Cls. acc. |
|---|---|---|
| CLS emission, temperature = 0.75 | 0.353 | 0.448 |
| CLS emission, temperature = 1.0 | 0.352 | 0.451 |
| CLS emission, temperature = 2.0 | 0.351 | 0.452 |
| Selected REG likelihood | 0.353 | 0.457 |
| Combined REG–CLS emission | **0.355** | 0.454 |

*Table 12.* **Reference-depth diagnostic at 1B samples.** Values are Aggregate Rank@1.

| $d_{ref}$ | 0 | 1 | 2 | 6 |
|---|---|---|---|---|
| Agg. Rank@1 | 0.365 | 0.384 | 0.400 | 0.407 |

narrows as the model sees more data. We retain it because it improves intermediate-depth behavior and does not produce a consistent degradation.

## H.5. EMA Start

The EMA teacher generates emissions for HMM refinement. Starting EMA emissions immediately can be unstable because the teacher is not yet informative. Table 13 reports a small-scale diagnostic at 250M samples. Delaying the start improves early stability. After roughly 1B samples, the differences fall below 0.002 Aggregate Rank@1, or about 0.2 percentage points.

*Table 13.* **EMA-start diagnostic at 250M samples.** Values are Aggregate Rank@1.

| EMA emission start | Agg. Rank@1 |
|---|---|
| 0 samples | < 0.300 |
| 20M samples | 0.363 |
| 50M samples | 0.358 |

