# OpenReview forum: "Trajectory-Aware Heuristic Learning for Combinatorial Search"
_ICML.cc/2026/Conference — ICML 2026 regular_

### Official Review · Reviewer_U9FS · 2026-02-25

**Soundness:** 2
**Presentation:** 1
**Significance:** 3
**Originality:** 3
**Overall Recommendation:** 4
**Confidence:** 3

**Summary:**

This paper proposes a framework for learning and evaluating heuristics for combinatorial problems where only a small set of exact optimal depths are available, such as the Rubik's cube. The framework treats depths along a trajectory as a latent sequence, using an anchored HMM to refine that combines sparse exact anchors and depth likelihoods from a teacher network. This yields smooth depth distributions, which a student model is trained on. The student model can then be used for down-stream search tasks. It also introduces CubeLocalRank, a benchmark that evaluates how well a heuristic ranks nearby successor states without running full search. Experiments show the resulting heuristic improves local ranking quality and solution quality on Rubik's cube benchmarks.

**Compliance With Llm Reviewing Policy:**

Affirmed.

**Final Justification:**

My main concerns about the paper have been addressed in the rebuttal response.

**Key Questions For Authors:**

1. For Table 1 (and any key downstream results), can you provide expansion counts, number of unique states visited, and wall clock time? This would resolve whether improvements to solution quality come via a larger computation cost.
2. Can you share any data about the training cost? In Appendix D you state that its similar to baselines but specifics would help here, as well as if this include data generation time (BFS to find the initial trajectories for $d \le d\_\text{ref}$)
3. For ablations C3, C4 and C5, claims are made but I can't find what results data they are referring to. Can you help clarify what is being used to justify these claims?
4. Is it possible to adapt your framework to environments which have deadlocks like Sokoban, where a single move could transition from a state with a low heuristic value to a state which can no longer have trajectories though it that lead to the goal? This transition no longer follows the $\{-1,0,1\}$ change in heuristic value.

**Limitations:**

yes

**Strengths And Weaknesses:**

**Strengths**
- This work touches on important ideas, being how to strengthen the supervision signal while enforcing heuristic consistency without relying entirely on optimal labeled trajectories
- I believe the framing of casting heuristic learning as probabilistic inference over trajectories via anchored HMM smoothing rather than per-state regression is novel
- The search-free heuristic quality benchmark is useful as a tool for rapid iteration without having to repeatedly run expensive downstream search
- Empirical results show an improvement over the main baseline in both the search-free and search benchmarks

**Weaknesses**

I feel that while this paper has a lot of good ideas, it needs quite a bit of review.
- Missing important decision-relevant metrics of training overhead (its talked about but not quantified) and search effort in terms of expansions and wall-clock time, so its hard to tell where this method lies on the quality/compute frontier
- There are several presentation issues that obstruct understanding
    - Figure 1 has the selection box highlighted on $E_t(\cdot)$, and grid background
    - Figure 2 has incorrect letters when referring to the sub-figures, and is visually too small
    - Missing space between "algorithms,particularly" online 260
    - The concepts of Level-1 and level-2 neighbors are mentioned but never defined anywhere
- You have several ablations which have a discussion about different tradeoffs, but some are missing data/results to justify the claims

---

> ### Author Rebuttal · Authors · 2026-03-30
>
> ## Q1. Search budget
>
> We thank the reviewer. This overlaps with Reviewer Awyg’s Q2; we restate the key evidence here.
>
> All: QTM, 1k DeepCubeA (avg)
>
> For beam search, we match the beam size and search procedure. At beam $2^{18}$:
>
> - **CayleyPy (our run):** 12.2s, 4.24M expansions
> - **Ours (without intermediate matching):** 12.1s, 4.21M expansions
>
> Thus, the search effort is matched, so the gain is not due to increased compute.
>
> With intermediate matching enabled, our method reduces redundant expansions **within the same beam constraint**, improving runtime ($12.2\text{s}\rightarrow 7.14\text{s}$) and reducing expansions. This is an efficiency gain under the same budget, not a larger search budget.
>
> For batch A*:
>
> - **DeepCubeA:** 75.6s, 8.2M expansions
> - **Ours:** 45.4s, 5.4M expansions
>
> We will add these metrics in the final version.
>
> ## Q2. Training cost
>
> We thank the reviewer. This overlaps with Reviewer Awyg’s Q3; we summarize the quantitative results here.
>
> The exact-depth store is a one-time preprocessing step (e.g., for the $3 \times 3 \times 3$ Cube in the QTM setting with $d_{\text{ref}}=7$, $\sim$37s and $\sim$721MB) and is not part of the per-iteration training cost.
>
> Under a matched 1M-sample setup:
>
> - **CayleyPy:** total $=3.11\text{s}$, train $=2.14\text{s}$
> - **Ours ($d_{\mathrm{ref}}=1$ to $7$):** total $=3.42\text{s}$ $\rightarrow$ $3.18\text{s}$, train $\approx 2.30\text{s}$
>
> The higher cost corresponds to $d_{\mathrm{ref}}=1$, while $d_{\mathrm{ref}}=7$ is faster since more states are resolved by anchors (fewer EMA emissions).
>
> With probing ($d_{\mathrm{ref}}=7$): probe$=1$ $\sim +4$%, probe$=2$ $\sim +13$%. We will add these metrics in the final version.
>
> ## Q3. Clarification for C3, C4, C5
> We thank the reviewer. We agree the appendix presentation was too compressed; the ablations were summarized textually to avoid many tables. We clarify the supporting results below.
>
> **C3. Reference depth ($d_{\mathrm{ref}}$).**
> The effect is consistent: moving from $d_{\mathrm{ref}}=0$ to $1$ gives a clear gain, while gains beyond $d_{\mathrm{ref}}\approx 2$ mostly saturate. For example, at $\sim 1$B samples, Rank@1 improves from $\sim 36.5$% at $d_{\mathrm{ref}}=0$ to $\sim 38.4$% at $d_{\mathrm{ref}}=1$, $\sim 40.0$% at $d_{\mathrm{ref}}=2$, and only $\sim 40.7$% at $d_{\mathrm{ref}}=6$. At larger scale ($\sim 15$B), $d_{\mathrm{ref}}=0$ remains weaker ($\sim 37$)%, while $d_{\mathrm{ref}}=1 -6$ is around $\sim 42$%. This is the basis for the C3 claim: shallow exact anchors matter, but deeper prior construction alone saturates; As shown in Fig. 2a, depth-change dynamics become similar beyond shallow depths, so at scale improvements come from training rather than prior construction.
>
> **C4. Probing**
>
> Probing extends supervision beyond $d_{\text{ref}}$ without increasing it.
>
> At $d_{\text{ref}}=6$ (no probe / probe1 / probe2), overall Rank@1 changes are small ($\sim$0.1%) and fluctuate, but consistent gains appear at deeper depths (e.g., $d \geq 7,8,9$), though they shrink with scale.
>
> At very large depths (e.g., $d \geq 15$), results become noisy because the state space grows exponentially and only a small fraction is observed during training.
>
> Thus, probing mainly improves deeper-depth ranking with little effect on aggregate Rank@1. We include it for scaling: shallow depths are well covered, but deeper depths are not—even at large scale—and this gap increases for larger puzzles.
>
> **C5. Optimization stability (EMA start)**
>
> At 250M samples, starting EMA emissions too early is unstable:
>
> - start = 0 $\rightarrow$ $<30$%
> - start = 20M $\rightarrow$ 36.3%
> - start = 50M $\rightarrow$ 35.8%
>
> Delaying EMA stabilizes early training; the exact start mainly affects early behavior. At $\geq 1$B samples, differences vanish ($<0.2\%$, within $\pm 0.2\%$ across seeds).
>
> We will revise the appendix/final version to make these links explicit.
>
> ## Q4. Deadlocks (e.g., Sokoban)
>
> We thank the reviewer for this important question.
>
> In prior value-learning work such as DeepCubeA, deadlocks are not modeled separately; such states are effectively assigned larger cost-to-go, and the method remains effective. Our framework follows the same principle: a transition into a dead-end state behaves as a depth-increasing transition, so it does not require special treatment in the current formulation.
>
> More generally, our method models probabilistic depth evolution rather than enforcing a strict deterministic $\pm 1$ change at every step. This allows dead-end transitions to be absorbed as high-cost outcomes without changing the basic training or search setup. Puzzle-specific extensions, such as explicit deadlock detection, could be added, but they are not required for the method to function or for comparison to prior value-based approaches.
>
> We will clarify this point in the final version.
>
> ---
> **Presentation**: We will revise the figures for clarity and correctness, addressing formatting and readability issues.

---

> > ### Author Rebuttal · Reviewer_U9FS · 2026-04-03
> >
> > Thank you for the response to the rebuttal, my concerns have been addressed. I have revised by score.

---

### Official Review · Reviewer_Awyg · 2026-03-11

**Soundness:** 2
**Presentation:** 3
**Significance:** 3
**Originality:** 3
**Overall Recommendation:** 4
**Confidence:** 3

**Summary:**

This paper studies heuristic learning for combinatorial search under noisy supervision from random walks. The main idea is to model depth along a trajectory as a latent variable and refine pseudo-labels using an anchored HMM-style procedure that combines shallow exact-depth anchors, a depth-dynamics prior, and neural emissions from a teacher model. The paper also proposes CubeLocalRank, a search-free local ranking benchmark intended to measure heuristic quality without relying solely on downstream search performance. Experiments on Rubik’s Cube in HTM/QTM settings and on larger puzzles show improved local ranking metrics and competitive downstream search performance.

**Compliance With Llm Reviewing Policy:**

Affirmed.

**Final Justification:**

The authors’ responses have addressed my main concerns and I appreciated their hard working during the rebuttal phase. I revise the score accordingly.

**Key Questions For Authors:**

1. Can the authors better isolate the marginal contribution of the anchored HMM refinement relative to simpler pseudo-labeling or smoothing alternatives under matched settings?
2. Can the authors clarify how search budgets are matched across different search algorithms/configurations, especially in the QTM comparisons?
3. How expensive is the HMM-based refinement during training in terms of wall-clock and memory overhead?
4. Do the authors have evidence that CubeLocalRank, or an approximate variant of it, remains predictive in domains without access to solver-labeled local neighborhoods?

**Limitations:**

Yes

**Strengths And Weaknesses:**

### Strengths

The paper is well motivated and addresses a real issue in heuristic learning, namely that step-based supervision from random walks is noisy and becomes less reliable at greater depths. The trajectory-level probabilistic labeling framework is technically interesting, and CubeLocalRank is a useful attempt to separate heuristic quality from search budget and implementation details. The empirical section is fairly extensive, with scaling results across training sizes and beam budgets, and the HTM results under matched settings appear meaningfully stronger than the reproduced baseline.

### Weaknesses

My main concerns are about attribution, comparability, and generality. First, it is still not fully clear how much of the gain comes specifically from the anchored HMM refinement versus other components such as consistency regularization and training heuristics. Second, some comparisons, especially under QTM, involve different search procedures or configurations, which makes strict matched-budget interpretation harder. Third, the broader significance of CubeLocalRank is not yet fully established beyond puzzle domains with access to solver-labeled local neighborhoods. These issues do not invalidate the paper, but they reduce my confidence in the strength and generality of the claims.

---

> ### Author Rebuttal · Authors · 2026-03-29
>
> ## **Q1. Marginal contribution of anchored HMM refinement**
>
> We thank the reviewer. We compare simplified pseudo-labeling variants under matched settings to isolate the anchored HMM.
>
> At 4.2B samples:
>
> - **Step-based supervision (CayleyPy-style) + anchors:** 39.9%
> - **Prior + anchors (no emissions):** 40.4%
> - **Prior-based random pseudo-labeling:** 38.1%
> - **Full pipeline (anchored HMM):** 41.3%
>
> The *prior + anchors* variant still uses HMM inference but replaces emissions with a uniform distribution. The random pseudo-labeling variant assigns labels independently from the step prior, without trajectory inference.
>
> The anchored HMM improves over both baselines ($+0.9\%$, $+3.2\%$), showing that gains come from trajectory-level inference combining prior, anchors, and learned emissions.
>
> At smaller scales (e.g., $0.25$B), differences are minimal ($<0.1\%$).
>
> ## **Q2. Search budget matching**
>
> We thank the reviewer for requesting clarification on search budget comparability.
>
> **Beam search (Table 1, QTM).**
> All comparisons at a given beam size use the same beam size and identical search procedure (including expansion rules and stagnation handling). In particular, our method with $d_{\text{ref}} = 0$ is algorithmically equivalent to CayleyPy, providing a strictly matched baseline.
>
> Results: 1k DeepCubeA instances; Table 1 models; avg per sample
>
> At beam size $2^{18}$:
>
> - **CayleyPy (our run):** 12.2s, 4.24M expansions
> - **Ours ($d_{\text{ref}} = 0$):** 12.1s, 4.21M expansions
>
> This confirms that search effort is matched, and improvements are not due to increased compute.
>
> For $d_{\text{ref}} > 0$, intermediate matching reduces redundant expansions, leading to lower runtime (12.1s → 7.14s) and fewer node expansions. This reflects improved efficiency within the same beam constraint, rather than an increased search budget.
>
> **Batch A* (DeepCubeA setting).**
>
> - **DeepCubeA:** 75.6s, 8.2M expansions
> - **Ours:** 45.4s, 5.4M expansions
>
> These results show that, across both beam search and A*, improvements arise from stronger heuristic guidance, not increased search budget. We will include these details in the final version.
>
> ## **Q3. Cost of HMM-based refinement**
>
> We thank the reviewer for asking about the training cost and overhead of the HMM-based refinement.
>
> *(Cube $3 \times 3 \times 3$, QTM setting.)*
>
> **(1) One-time preprocessing (exact-depth store).**
> The store is constructed once via BFS for depths $d \leq d_{\text{ref}}$, and its size grows exponentially:
>
> - Depth 4: $\sim$10k states
> - Depth 5: $\sim$94k
> - Depth 6: $\sim$0.88M
> - Depth 7: $\sim$8.2M
>
> In practice:
> - $d_{\text{ref}} = 7$: $\sim$37s preprocessing, $\sim$721 MB memory
>
> This is a one-time cost and does not affect training iterations.
>
> **(2) Per-iteration cost (including data generation, $\sim$1M samples).**
> We measure cost under an exactly matched setup with CayleyPy-style training:
>
> - **Baseline (CayleyPy-style):**
>   data + train = 3.11s (train $\approx$ 2.14s)
>
> - **Ours:**
>   $d_{\text{ref}} = 1$: total = 3.42s
>   $d_{\text{ref}} = 7$: total = 3.18s (train $\approx$ 2.30s)
>
> This shows that:
> (i) the training step is slightly higher ($\approx$2.14s → $\approx$2.30s) due to additional losses;
> (ii) the total cost remains comparable, and even decreases with larger $d_{\text{ref}}$ as more states are resolved via anchors, reducing model evaluations during data generation.
>
> Overall, this corresponds to a $\sim$2–3% overhead at $d_{\text{ref}} = 7$.
>
> **(3) Effect of probing.**
> When using probing (to extend supervision beyond feasible $d_{\text{ref}}$):
>
> - probe = 1: total = 3.24s ($\sim$+4%)
> - probe = 2: total = 3.51s ($\sim$+13%)
>
> Thus, probing introduces additional cost, but remains moderate.
>
> **Summary.**
> Preprocessing is one-time, training cost is slightly higher, and total per-iteration cost remains comparable, with $\sim$2–3% overhead (up to $\sim$10–13% with probing). We will include these details in the final version.
>
> ## **Q4. Generality of CubeLocalRank beyond solver-labeled domains**
>
> We agree that CubeLocalRank relies on solver-labeled local neighborhoods and is therefore most directly applicable to domains like $3 \times 3 \times 3$ where exact supervision is available.
>
> However, CubeLocalRank is not required by the framework. It is a search-free diagnostic used to evaluate heuristic quality and guide learning design without expensive downstream search; the training method itself does not depend on it.
>
> In this sense, it serves as a development platform: strategies validated in this setting (e.g., trajectory-based supervision, consistency constraints) transfer to domains without exact supervision, as reflected in our larger-puzzle results.
>
> Finally, approximate variants are feasible: local neighborhoods can be constructed via limited search or partial supervision, where preserving relative ordering is sufficient.
>
> We will clarify this positioning in the final version.

---

> > ### Author Rebuttal · Reviewer_Awyg · 2026-04-02
> >
> > My concerns have been addressed. But I still maintain the score.

---

### Official Review · Reviewer_7t1G · 2026-03-12

**Soundness:** 2
**Presentation:** 3
**Significance:** 2
**Originality:** 3
**Overall Recommendation:** 4
**Confidence:** 2

**Summary:**

This paper's main idea concerns learning better value heuristics for combinatorial search without costly search-in-the-loop training. A critical issue assessed by the study is that random-walk-generated training labels systematically overestimate true optimal depth, producing noisy supervision. To address this, the authors model depth as a latent trajectory variable and apply anchored HMM inference to refine noisy labels into consistent pseudo-labels, supplemented by a neighbor-consistency loss during training. They also introduce CubeLocalRank, a search-free benchmark for evaluating heuristic quality through local state ranking. Experiments on the Rubik's Cube show consistent improvements in solution optimality and solvability over strong baselines under matched computational budgets.

**Compliance With Llm Reviewing Policy:**

Affirmed.

**Final Justification:**

The authors’ responses helped address my main concerns and clarified several aspects of the work.

**Key Questions For Authors:**

- The neighbor-consistency loss is shown to be the most impactful training component, but it is unclear how much of this benefit depends on the specific single-step trajectory continuation strategy. Would similar gains be observed if trajectories were generated with goal-rooted restarts as in CayleyPy?

**Limitations:**

Yes.

**Strengths And Weaknesses:**

Strengths
- The probabilistic label refinement via anchored HMM is a principled and novel way to address the noisy supervision problem inherent in random-walk training data.
- CubeLocalRank provides a practical, search-free diagnostic that decouples heuristic quality evaluation from expensive downstream search.

Weaknesses
- CubeLocalRank requires solver-labeled local neighborhoods, which may be unavailable or expensive in domains without efficient optimal solvers.
- Scalability to larger puzzles such as 6×6×6 remains limited, with only partial results reported due to computational constraints.

---

> ### Author Rebuttal · Authors · 2026-03-29
>
> We thank the reviewer for the constructive feedback.
>
> **CubeLocalRank.**
> CubeLocalRank is a concrete search-free benchmark instantiating our local-ordering evaluation protocol in a setting where exact local supervision is available. However, CubeLocalRank is **not required** by the proposed framework. It is introduced as a search-free diagnostic to evaluate heuristic quality and guide learning design without expensive downstream search; the training method itself does not depend on it. In this sense, CubeLocalRank serves as a development platform: strategies validated in this controlled setting (e.g., trajectory-based supervision, consistency constraints) can then be transferred to domains without exact local supervision, as reflected in our larger-puzzle results. Approximate variants are also feasible, where local neighborhoods are constructed using limited search or partial supervision and only relative ordering is needed. We will clarify this positioning in the final version.
>
> **Scalability.**
> We agree that results on $6 \times 6 \times 6$ are limited by computational constraints and should be viewed as preliminary. However, results on larger domains already show encouraging scaling: on $4 \times 4 \times 4$ we achieve full solvability with a single model, and on $5 \times 5 \times 5$, to our knowledge, we are the first to solve the full Santa benchmark with a single model, whereas prior work (e.g., CayleyPy) relies on ensembles at this scale. This suggests that our approach scales competitively under practical budgets. We therefore view $6 \times 6 \times 6$ as a compute-limited frontier shared across methods, rather than evidence against scalability.
>
> **Neighbor consistency and trajectory strategy.**
> We agree this is an important question. The key issue is that our method is built around **temporally linked trajectories** generated by **single-step continuation**, whereas CayleyPy-style goal-rooted sampling produces **independent restart samples**. As described in the paper, our refinement stage requires genuinely connected trajectories; thus a direct comparison of the *full pipeline* under goal-rooted restarts is not applicable.
>
> We therefore isolate the effect of **neighbor consistency** under both sampling strategies at the same training scale (models trained on 10B samples, unless noted otherwise):
>
> 1. **Goal-rooted, no neighbor consistency (CayleyPy, 10B):**
>    **40.0 Rank@1 / 59.6% accuracy**.
>
> 2. **Goal-rooted, with neighbor consistency:**
>    because restart-based samples are independent, we approximate neighbor consistency by pairing each state with a true successor and applying the same local consistency loss. This effectively doubles supervision, so **10B restart samples correspond to 20B effective state samples**. This improves performance to **41.4 / 62.9**.
>
> 3. **Single-step trajectories, no neighbor consistency (10B):**
>    using our trajectory construction but removing neighbor consistency gives **38.2 / 58.1**, showing that temporal linkage alone does not explain the gain.
>
> 4. **Proposed method (single-step trajectories + neighbor consistency + refinement):**
>    our full model achieves **42.2 / 63.3** at **10B**, and **42.6 / 66.6** at **22B**.
>
> Overall, the answer is: **yes, local-consistency gains are still present under goal-rooted restart-style sampling**, but the effect is **stronger and more natural with true connected trajectories**. This is precisely the regime targeted by our method, since both the anchored HMM refinement and the neighbor-consistency objective are designed for temporally linked trajectories. We will revise the final version to make this distinction explicit.

---

> > ### Author Rebuttal · Reviewer_7t1G · 2026-04-04
> >
> > My main concerns have been addressed, and I have updated my score accordingly.

---

### Official Review · Reviewer_WfsA · 2026-03-24

**Soundness:** 2
**Presentation:** 3
**Significance:** 3
**Originality:** 3
**Overall Recommendation:** 4
**Confidence:** 3

**Summary:**

The paper tackles the problem of learning heuristics for combinatorial optimization. The work proposes a framework for heuristic that combines trajectory based with supervision. The work introduces a new dataset CubeLocalRank for evaluating heuristics, and provides empirical results for Rubik's cube puzzle. Empirical results indicate that the heuristics developed achieve better performance.

**Compliance With Llm Reviewing Policy:**

Affirmed.

**Key Questions For Authors:**

1. Is it possible to obtain optimal solutions for bigger cubes like 4x4x4 etc, if so what do you think would be necessary?
2. What is the bottleneck of this approach? Do you think this can scale to 11x11x11 cube?
3. Mainly the Rubik's cube problem is considered, along with Global 6x10 puzzle, why was this particular puzzle chosen?

Answering these questions would improve my understandability of this paper.

**Limitations:**

Yes.

**Strengths And Weaknesses:**

Soundness: The claims all seem to be well supported. Empirical results also seem to be well designed by constructing Abelation studies However, the problem size bottleneck is not yet clear.
Presentation: Adding an example to describe the methodology would be really helpful. I think Figure 1 does a good job of explaining, the overall procedure. Please fix E_t on the arrow from the teacher. It has some issue with a box highlighted around it.
Significance: The paper does address an important problem, and could influence future research.
Originality: The work introduces new method along with data.

---

> ### Author Rebuttal · Authors · 2026-03-30
>
> We thank the reviewer for the positive feedback and helpful questions.
>
> **Q1. Optimal solutions for larger cubes (e.g., $4 \times 4 \times 4$)**
>
> For cubes beyond $3 \times 3 \times 3$, optimal solutions are generally not available in practice. While exact methods such as IDA* can be applied, they become prohibitively expensive as the state space grows. In addition, learning-based heuristics are not admissible, so optimality cannot be guaranteed. Therefore, consistent with prior work, evaluation on larger cubes focuses on solvability and solution quality. In this setting, our method achieves strong results (e.g., full solvability on $5 \times 5 \times 5$), indicating improved heuristic quality even without optimal supervision.
>
> **Q2. Bottleneck and scalability**
>
> The main bottleneck is the rapid growth of the search space, which affects both learning and downstream inference. As cube size increases, the number of reachable states, effective depth range, branching structure, and search horizon all grow sharply. This makes supervision noisier and harder to model during training, while also making downstream search increasingly expensive.
>
> In our framework, the dominant practical bottleneck is therefore **computational tractability**: for larger cubes, both maintaining sufficient coverage of the induced depth space during training and running strong downstream search become progressively more costly. This is also why our largest-scale results should be interpreted under fixed compute budgets and comparable evaluation protocols.
>
> Our results already show strong scaling through $5\times5\times5$, while $6\times6\times6$ remains compute-limited in the current study. We therefore view larger settings such as $11\times11\times11$ not as conceptually out of scope, but as requiring substantially greater compute and likely additional problem-specific engineering to remain tractable under practical budgets.
>
> **Q3. Choice of puzzles**
>
> We use Rubik’s Cube as a canonical benchmark that allows controlled evaluation and scaling ($3 \times 3 \times 3 \rightarrow 5 \times 5 \times 5$), enabling both precise analysis (via local ranking) and large-scale search evaluation.
>
> To test generalization beyond cube domains, we also evaluate on the Globe $6 \times 10$ puzzle, which has a much larger state space ($\sim 10^{145}$). In addition, we train and evaluate on other permutation puzzles (e.g., Puzzle-35 and Pancake-45) to verify that the approach is not specific to cube structure. Across these settings, the learned heuristic transfers effectively, demonstrating the generality of the framework.
>
> **Presentation**
>
> We will fix the visualization issue noted in Figure 1 and improve clarity with a brief illustrative example in the final version.

---

> > ### Author Rebuttal · Reviewer_WfsA · 2026-04-06
> >
> > My concerns have been addressed. But I still maintain the score.

---

### Decision · Program_Chairs · 2026-04-30

**Decision:**

Accept (regular)

**Comment:**

All reviewers agree on Weak Accept, but not going beyond, to Accept or Strong Accept.

Hence, I suggest Weak Accept.